# Split intein-mediated selection of cells containing two plasmids using a single antibiotic

Navaneethan Palanisamy [1,2,3], Anna Degen [2,4,5], Anna Morath[1,6,7,8], Jara Ballestin Ballestin [1,2], Claudia Juraske[1,6,8], Mehmet Ali Öztürk[1,2], Georg A. Sprenger [9], Jung-Won Youn [9], Wolfgang W. Schamel[1,6,8] & Barbara Di Ventura [1,2]*

To build or dissect complex pathways in bacteria and mammalian cells, it is often necessary to recur to at least two plasmids, for instance harboring orthogonal inducible promoters. Here we present SiMPl, a method based on rationally designed split enzymes and intein-mediated protein *trans*-splicing, allowing the selection of cells carrying two plasmids with a single antibiotic. We show that, compared to the traditional method based on two antibiotics, SiMPl increases the production of the antimicrobial non-ribosomal peptide indigoidine and the non-proteinogenic aromatic amino acid *para*-amino-L-phenylalanine from bacteria. Using a human T cell line, we employ SiMPl to obtain a highly pure population of cells double positive for the two chains of the T cell receptor, TCRα and TCRβ, using a single antibiotic. SiMPl has profound implications for metabolic engineering and for constructing complex synthetic circuits in bacteria and mammalian cells.

---

[1] Signalling Research Centres BIOSS and CIBSS, University of Freiburg, 79104 Freiburg, Germany. [2] Institute of Biology II, Faculty of Biology, University of Freiburg, 79104 Freiburg, Germany. [3] Heidelberg Biosciences International Graduate School (HBIGS), 69120 Heidelberg, Germany. [4] BioQuant Center for Quantitative Biology, University of Heidelberg, 69120 Heidelberg, Germany. [5] DKFZ Graduate School, University of Heidelberg, 69120 Heidelberg, Germany. [6] Department of Immunology, Institute of Biology III, Faculty of Biology, University of Freiburg, 79104 Freiburg, Germany. [7] Spemann Graduate School of Biology and Medicine (SGBM), University of Freiburg, 79104 Freiburg, Germany. [8] Center for Chronic Immunodeficiency (CCI), Medical Center Freiburg and Faculty of Medicine, University of Freiburg, 79104 Freiburg, Germany. [9] Institute of Microbiology, University of Stuttgart, 70569 Stuttgart, Germany. *email: barbara.diventura@biologie.uni-freiburg.de

Inteins are proteins that excise themselves out of their host proteins in an autocatalytic process called protein splicing, which leads to the formation of a new peptide bond between the two polypeptides originally flanking the intein[1,2]. Inteins can be naturally or artificially split into two fragments that carry out a *trans*-splicing reaction[3,4] (Fig. 1a). For their ability to make fusions between two previously separate polypeptides, split inteins can be used to reconstitute a protein out of two dysfunctional parts[5–7]. Protein functional complementation may be alternatively achieved by bringing the dysfunctional parts into close physical proximity without formation of a peptide bond, as done by the inteins. This method, routinely applied for example to study protein–protein interactions[8–10], was used to prove the principle that bacterial cells carrying two or three plasmids could be selected using a single antibiotic[11]. In the two plasmids version of the system, aminoglycoside 3′-phosphotransferase, the enzyme conferring resistance towards kanamycin (for simplicity, in the following referred to as APT), is split into two fragments. Each fragment is fused to a leucine zipper and cloned in a separate plasmid under an isopropyl β-D-1—thiogalactopyranoside (IPTG)-inducible promoter. The fragments come in close proximity via the leucine zippers, regaining activity and conferring kanamycin resistance only to cells containing both plasmids. In the three plasmids-version of the system, the arabinose-inducible pBAD promoter controls the expression of the T7 RNA polymerase, which in turn leads to the production of the two fragments of the enzyme fused to leucine zippers, cloned under the

pT7/lac promoter. Each component of the system is present on a separate plasmid. Thus, only cells containing all three plasmids are resistant to kanamycin. In both variants of the method, every plasmid contains an additional resistance cassette. This approach has one major drawback: the genes belonging to the circuitry conferring resistance (T7 polymerase and/or two enzyme fragments) are cloned under inducible promoters in the multiple cloning site (MCS). However, typically, it is desirable to employ the inducible promoters to control the expression of genes of interest rather than the components of the resistance cassette. This hinders the usage of the system for real applications. Moreover, the method was applied only to the previously split APT[12] and was limited to bacteria. The possibility to select plant and mammalian cells containing two plasmids with a single antibiotic would be beneficial for basic as well as applied science, because there are not many antibiotics that work well in these model systems[13,14].

Here we describe SiMPl, a method based on split intein-mediated enzyme reconstitution, that allows selecting bacterial and mammalian cells containing two plasmids with a single antibiotic (Fig. 1b). SiMPl is implemented through plasmids, the SiMPl plasmids (Fig. 2a, b, Supplementary Fig. 1 and Supplementary Fig. 2), which can be readily used to regulate the expression of target genes in bacterial and mammalian cells, as the split resistance cassette simply substitutes the full-length cassette. Using computational approaches, we split the enzymes conferring resistance towards ampicillin, chloramphenicol,

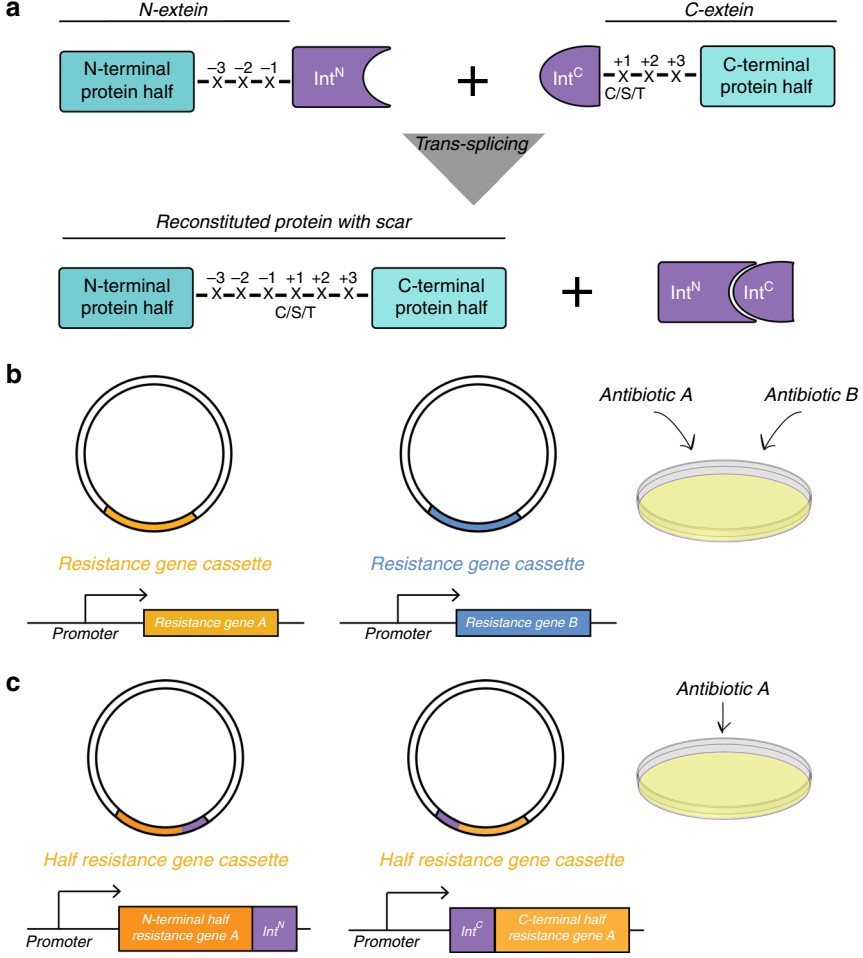

**Fig. 1** Cells containing two plasmids can be selected with a single antibiotic. **a** Schematic showing *trans*-splicing of proteins by a split intein. X any amino acid, C cysteine, S serine, T threonine. **b**, **c** Schematic highlighting the difference between the conventional (**b**) and the SiMPl (**c**) selection methods

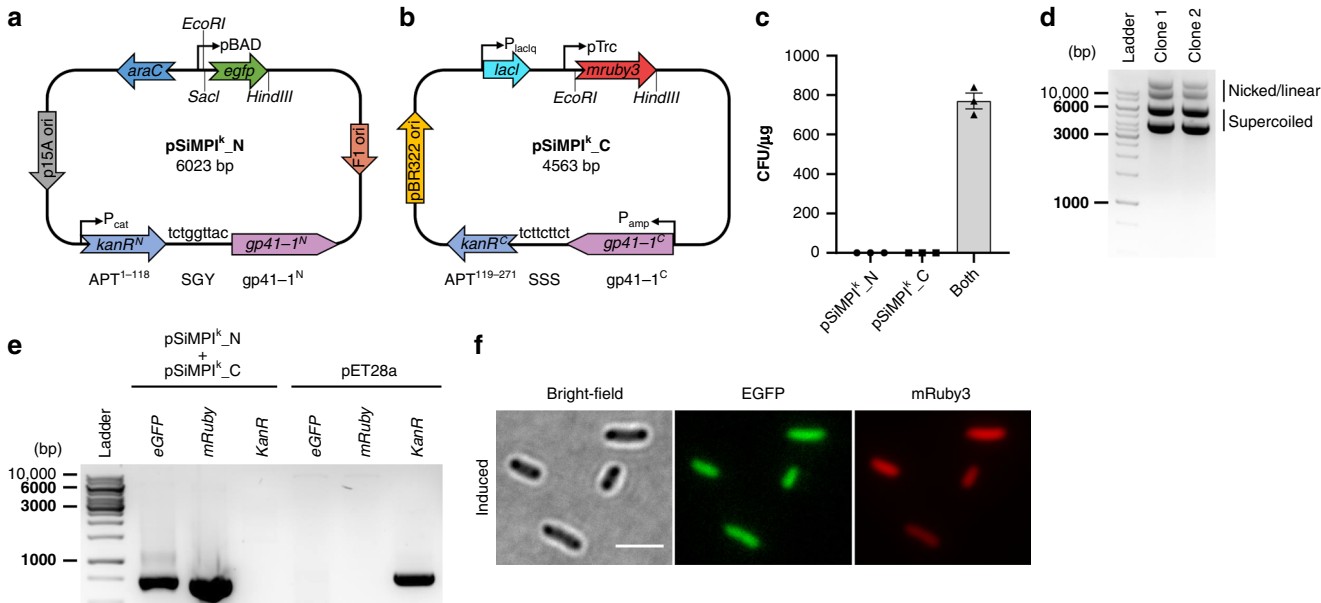

**Fig. 2** The SiMPl method based on kanamycin. **a**, **b** Schematic showing the main features found on the SiMPl plasmids. pSiMPl$^k$_N (**a**) and pSiMPl$^k$_C (**b**) are derivatives of pBAD33 and pTrc99a, respectively. The chloramphenicol resistance gene in pBAD33 is replaced by a fragment of the kanamycin resistance gene encoding amino acids 1 to 118 of aminoglycoside 3'-phosphotransferase followed by the N-terminal fragment of the split gp41-1 intein. Similarly, the ampicillin resistance gene in pTrc99a is replaced by the C-terminal fragment of gp41-1 followed by a fragment of the kanamycin resistance gene encoding amino acids 119 to 271 of aminoglycoside 3'-phosphotransferase. For more efficient splicing, the local exteins (SGY and SSS) are included. **c** Bar graph showing the transformation efficiency in *E. coli* TOP10 cells of the indicated plasmids. Values represent mean (± standard error of the mean) of three independent experiments. **d** Ethidium bromide-stained agarose gel showing plasmid DNA isolated from two randomly picked clones obtained after transformation of *E. coli* TOP10 cells with the SiMPl plasmids shown in (**a**) and (**b**). **e** PCR analysis of the SiMPl plasmids isolated from bacteria. pET28a was used as control to show the product obtained after amplification of the full-length kanamycin resistance gene. **f** Representative fluorescence microscopy images of *E. coli* TOP10 cells carrying the SiMPl plasmids shown in (**a**) and (**b**) induced with 0.1% arabinose and 1 mM IPTG for 3 h. Scale bar, 3 μm. Source data are provided as a Source Data file

hygromycin and puromycin. We show that, when using the SiMPl plasmids to express enzymes involved in a biosynthetic pathway, bacteria produce higher amounts of valuable chemicals. Furthermore, we demonstrate that SiMPl based on puromycin can be applied to obtain highly pure populations of mammalian cells double positive for two vectors of interest.

## Results

**SiMPl for selection with kanamycin.** To construct pSiMPl$^k$_N and pSiMPl$^k$_C, the two plasmid constituents of the SiMPl method based on kanamycin, we selected two commonly used backbones, pBAD33 and pTrc99a. pBAD33 allows inducible expression of a gene cloned in the MCS using arabinose and harbors the chloramphenicol resistance gene. pTrc99a allows inducible expression of a gene cloned in the MCS using IPTG and harbors the ampicillin resistance gene. The residue at which to split APT into two fragments was previously established[15]. As split intein we selected the extremely efficient gp41-1[16], which has serine as catalytic residue at position + 1 (Fig. 1a). We therefore included this residue upstream of the C-terminal fragment of APT (Fig. 2a). Moreover, to secure high efficiency of the splicing reaction, we decided to include five additional residues, three upstream of the N-terminal gp41-1 fragment ('SGY', at positions −3, −2, −1) and two downstream of the catalytic serine ('SS', at positions +2 and +3), since they represent the natural so-called local exteins for this intein[16] (Fig. 2a). We swapped the chloramphenicol resistance gene in pBAD33 with a fragment of the kanamycin resistance gene coding for residues 1 to 118 of APT followed by the gene coding for the N-terminal gp41-1 intein fragment (Fig. 2a). In the MCS, we cloned the *egfp* gene. Using

the same strategy, we swapped the ampicillin resistance gene in pTrc99a with the C-terminal gp41-1 intein fragment followed by a fragment of the kanamycin resistance gene coding for residues 119 to 271 of APT (Fig. 2b). In the MCS, we cloned the *mruby3* gene. We then transformed pSiMPl$^k$_N and pSiMPl$^k$_C either individually or together in *E. coli* TOP10 cells. Only cells co-transformed with both plasmids grew on the kanamycin-containing plates (Fig. 2c). Agarose gel electrophoretic analysis of the DNA extracted from two randomly-picked colonies indicated the presence of two plasmids (Fig. 2d). Polymerase chain reaction (PCR) confirmed the presence of the genes of interest (*egfp* and *mruby3*) and the absence of an intact kanamycin resistance gene (Fig. 2e). Finally, successful protein reconstitution was assessed by Western blotting (Supplementary Fig. 3). SiMPl$^k$_N and SiMPl$^k$_C can be used as the original pBAD33 and pTrc99a from which they are derived to control the expression of the genes of interest with two externally supplied inducers (Fig. 2f).

To understand if enzyme functionality required protein *trans*-splicing or could rely simply on physical proximity, we mutated two highly conserved residues in gp41-1 which are necessary for the *trans*-splicing reaction, namely the cysteine and the asparagine, first and last amino acids of gp41-1, respetively[17]. Interestingly, we found that mutation of the asparagine to alanine did not completely abolish cell growth in a medium containing kanamycin (Fig. 3a, b). To prove that the SiMPl method is advantageous in comparison to the traditional method based on two antibiotics, we quantified the transformation efficiency obtained with no plasmid, pBAD33 and pTcr99a, pSiMPl$^k$_N and pSiMPl$^k$_C (SiMPl$^k$) or pET28a, a plasmid carrying the

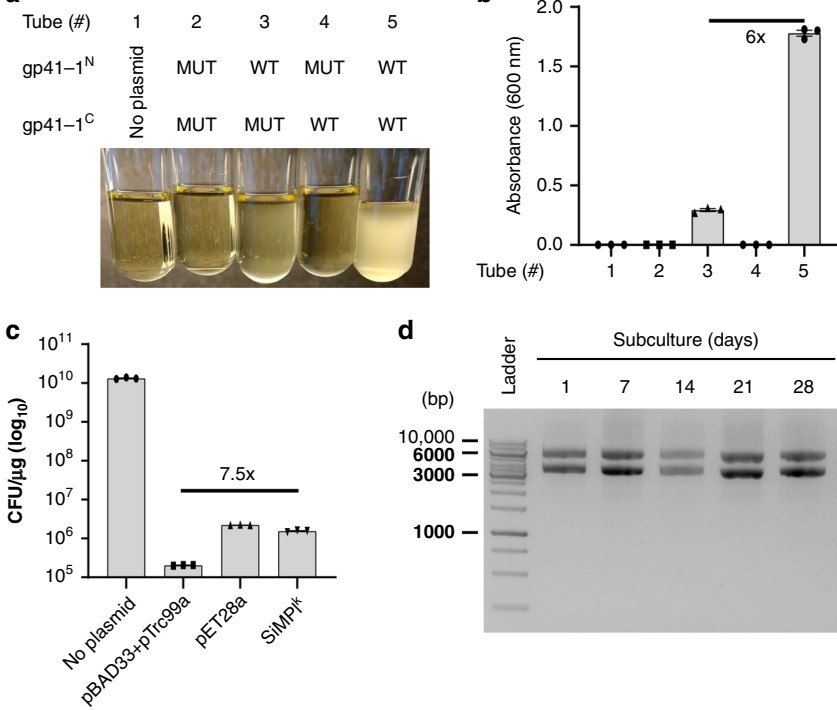

**Fig. 3** Characterization of SiMPl based on kanamycin. **a, b** gp41-1-mediated reconstitution of aminoglycoside 3′-phosphotransferase is needed to confer resistance to kanamycin. **a** Representative image of liquid cultures of *E. coli* TOP10 cells carrying either no plasmids (Tube # 1) or the SiMPl plasmids shown in Fig. 1 a and b (Tubes # 2-5), with (Tubes # 2-4) or without (Tube #5) the indicated mutations to gp41-1. gp41-1$^N$ MUT, mutation of the conserved cysteine at the very N-terminus of the N-terminal intein fragment to alanine; gp41-1$^C$ MUT, mutation of the conserved asparagine at the very C-terminus of the C-terminal intein fragment to alanine; WT, wild type. **b** Bar graph showing the values of the absorbance at 600 nm for the cultures in (**a**). Values represent mean (± standard error of the mean) of three independent experiments. **c** Transformation of SiMPl plasmids is more efficient than transformation of two classical plasmids carrying full-length resistance genes. Bar graph showing transformation efficiency in *E. coli* TOP10 cells of the indicated plasmids. For the "No plasmid" case, no antibiotic was applied to the plate. For all other cases, the appropriate antibiotics were added to the plates at a final concentration of 50 μg/mL for kanamycin, 100 μg/mL for ampicillin and 35 μg/mL for chloramphenicol. Values represent mean (± standard error of the mean) of three independent experiments. **d** SiMPl plasmids are maintained in bacteria. Ethidium bromide-stained agarose gel showing plasmid DNA isolated at the indicated time points from a culture of *E. coli* TOP10 cells transformed with the SiMPl plasmids based on kanamycin grown for a month. Source data are provided as a Source Data file

kanamycin resistance gene. With the SiMPl method, we obtained as many colonies as when transforming only one plasmid, while the double transformation selected on two antibiotics yielded 7.5 times fewer colonies (Fig. 3c). We also tested four different DNA concentrations in the transformation and compared the number of colonies obtained when transforming cells with only one plasmid (pET28a) or SiMPl$^k$ (Supplementary Fig. 4a). We found no difference in the number of colonies obtained with SiMPl$^k$ when using 100 and 10 ng of DNA. With 1 ng of DNA there were 4 times fewer colonies for SiMPl$^k$ compared to pET28a, however it should be noted that we are comparing cells transformed with two plasmids (SiMPl$^k$) with cells transformed with one plasmid (pET28a). Going down to 0.1 ng of DNA led to no colonies for SiMPl$^k$. To determine whether the cells would lose the SiMPl plasmids after some generations, we grew a bacterial culture for a month and took samples to extract plasmid DNA at regular intervals during this period. The SiMPl$^k$ plasmids were maintained throughout the entire period (Fig. 3d).

**SiMPl for selection with ampicillin and chloramphenicol**. To expand the SiMPl toolbox, we then sought to split and reconstitute other enzymes commonly used in bacteria, namely chloramphenicol acetyltransferase (CAT), for resistance towards chloramphenicol, and TEM-1 β-lactamase, for resistance towards ampicillin (in the following for simplicity referred to as TEM-1).

CAT, to the best of our knowledge, has never been split before. TEM-1 has been split before (at positions G194[18] and E195[19], where the numbering refers to the full-length protein); however, we were interested in whether our computational approach (see below) would identify the same residues.

In order to split the enzymes, we had to find the appropriate splice sites that would consent us to have two well-folded, yet dysfunctional protein fragments. The splice sites would additionally need to be located in regions of the protein allowing insertion of the six additional amino acids required for efficient splicing (Fig. 1a) without disruption of enzymatic activity. To locate the splice sites according to these requirements we used primarily information about the protein structure flexibility, albeit in some cases we additionally looked for sites of lower conservation (Supplementary Note 1). We reasoned that flexible regions likely tolerate better mutations or insertion of additional amino acids, even more so if these regions are additionally less conserved. Specifically, we simulated protein structure flexibility using the well-established, coarse-grained CABS-flex protein structure flexibility simulation tool[20] (Fig. 4a, b). In this model, the flexibility of the protein is given in terms of the fluctuations of the Cα atoms (root mean squared fluctuation or RMSF) of the protein backbone structural ensemble. Regions with higher RMSF are more flexible. To identify functionally important residues we applied the evolutionary trace method[21], which allowed us to analyze evolutionary patterns of sequence variations and locate

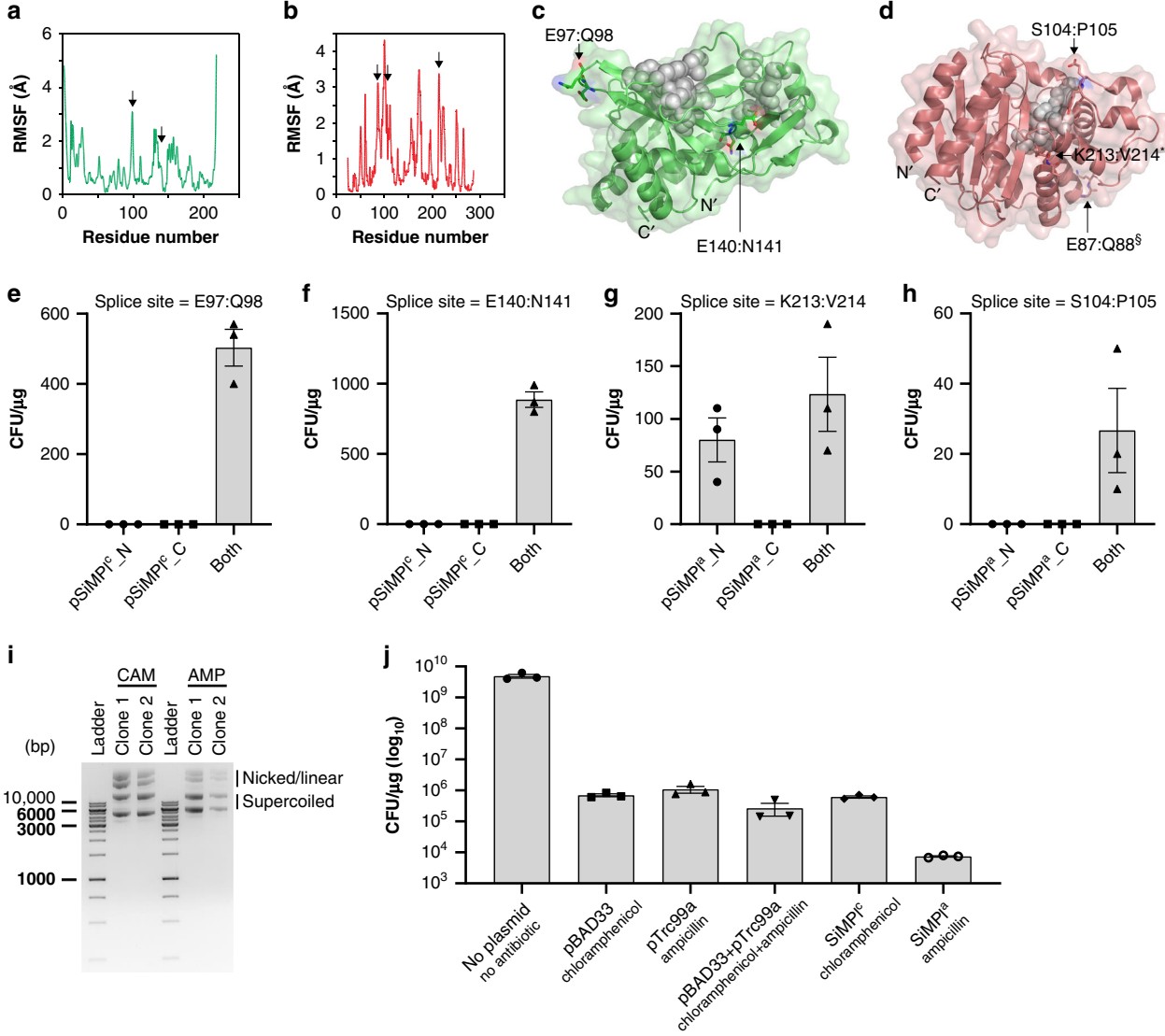

**Fig. 4** SiMPl based on chloramphenicol and ampicillin. **a**, **b** Root mean square fluctuation (RMSF) of Cα atoms in chloramphenicol acetyltransferase (PDB ID: 1q23) (**a**) and TEM-1 β-lactamase (PDB ID: 1zg4) (**b**) obtained from protein structure fluctuation simulations via the CABS-flex 2.0 web-server[20]. Flexible regions, within which splice sites were selected, are indicated by black arrows. **c**, **d** Crystal structure of TEM-1 β-lactamase (PDB ID: 1zg4) (**c**) and chloramphenicol acetyltransferase (PDB ID: 1q23) (**d**). Residues that are part of the active site are represented as gray spheres. Splice sites are represented as sticks and are pointed at by black arrows. Asterisk (*), splice site where the N-terminal fragment of the enzyme showed some independent antibiotic activity. Section sign (§), splice site that did not support bacterial growth. **e,h** Bar graph showing the transformation efficiency in *E. coli* TOP10 cells of the indicated plasmids. Values represent mean (± standard error of the mean) of three independent experiments. **i** Ethidium bromide-stained agarose gel showing plasmid DNA isolated from two randomly picked clones obtained after transformation of *E. coli* TOP10 cells with the SiMPl plasmids based on chloramphenicol (CAM) and ampicillin (AMP). **j** Bar graph showing the transformation efficiency in *E. coli* TOP10 cells of the indicated plasmids. Values represent mean (± standard error of the mean) of three independent experiments. Source data are provided as a Source Data file

them in relation to the splice sites (Supplementary Fig. 5). The residues we selected as splice sites were, additionally, surface-exposed and not located in the active site (Fig. 4c, d). Both splice sites for CAT supported bacterial growth exclusively when both plasmids were transformed (Fig. 4e, f), suggesting that they allow protein reconstitution to occur and that the reconstituted protein is functional. For TEM-1, splice site E87:Q88 did not support bacterial growth, while, interestingly, splice site K213:V214 led to a partially active protein fragment (the N-terminal one, from amino acid 1 to amino acid 213) (Fig. 4g). Splice site S104:P105 supported instead growth only when both plasmids were transformed (Fig. 4h). The previously used sites (G194:L196 and E195:L196), despite lying in a relatively flexible region

(Supplementary Fig. 6), were not among the best candidates according to our selection criteria.

As done for APT, we analyzed the need for intein-mediated *trans*-splicing for the activity of split CAT and TEM-1 using gp41-1 mutants. For CAT split between E97 and Q98, there was a 2.5 times reduction of cell growth when the conserved asparagine in the C-terminal gp41-1 intein fragment was mutated to alanine (Supplementary Fig. 7a). For TEM-1, interestingly, this mutation had no effect on cell growth (Supplementary Fig. 7c). For CAT split between E140 and N141, instead, all mutations nearly abolished bacterial growth (Supplementary Fig. 7b).

We selected splice site E140:N141 for CAT and splice site S104:P105 for TEM-1 to establish SiMPl^c_N and SiMPl^c_C (SiMPl^c),

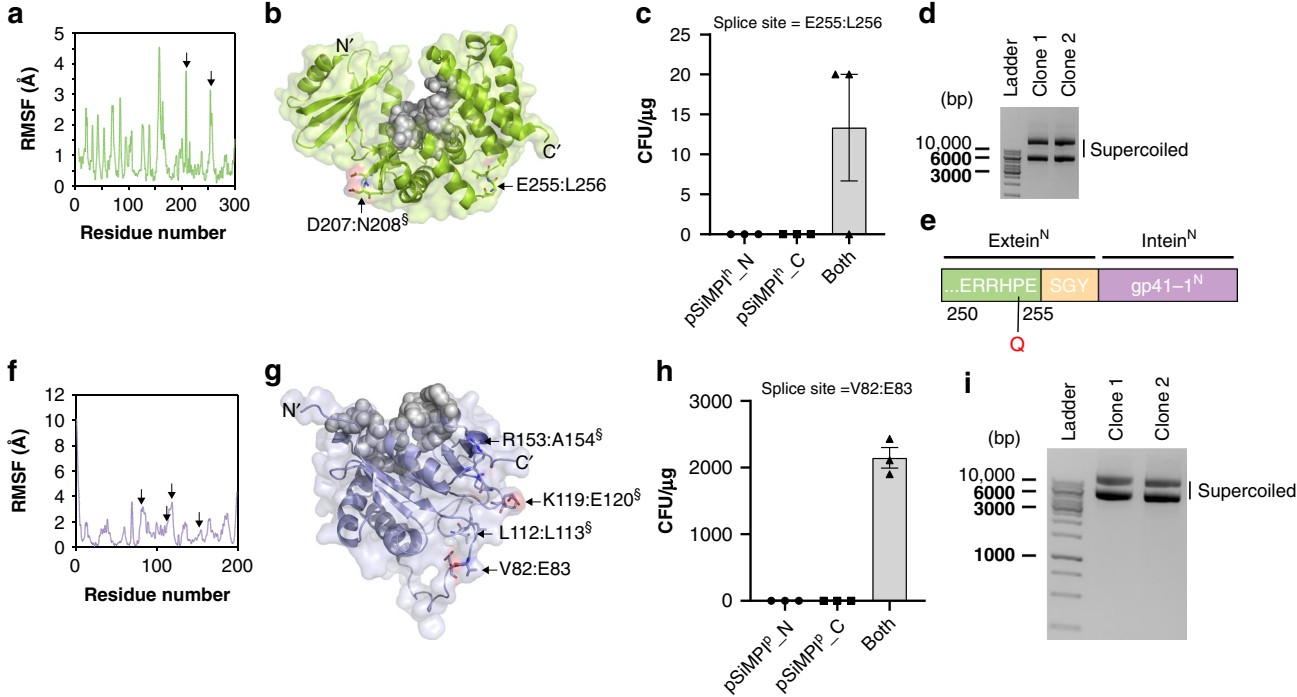

**Fig. 5** SiMPl based on hygromycin and puromycin. **a**, **f** Root mean square fluctuation (RMSF) of Cα atoms in hygromycin B phosphotransferase (PDB ID: 3w0s) (**a**) and the model structure of puromycin acetyltransferase (**f**) obtained from protein structure fluctuation simulations via the CABS-flex 2.0 web-server[20]. Flexible regions, within which splice sites were selected, are indicated by black arrows. **b** Crystal structure of hygromycin B phosphotransferase (PDB ID: 3w0s). **g** Tertiary structure model of puromycin acetyltransferase. (**b**) and (**g**) Residues that are part of the active site are represented as gray spheres. Splice sites are represented as sticks and are pointed at by black arrows. Section sign (§), splice site that did not support bacterial growth. **c**, **h** Bar graph showing the transformation efficiency in *E. coli* TOP10 cells of the indicated plasmids. Values represent mean (± standard error of the mean) of three independent experiments. **d**, **i** Ethidium bromide-stained agarose gel showing plasmid DNA isolated from two clones obtained after transformation with the SiMPl plasmids based on hygromycin (**d**) and puromycin (**i**). **e** Schematic representation of the N-terminal intein construct in pSiMP^h_N based on hygromycin. The last few residues of the N-terminal fragment of hygromycin B phosphotransferase are shown. The residue that was mutated in vivo by the bacteria is highlighted (white, WT residue; red, acquired residue). Source data are provided as a Source Data file

to use with chloramphenicol, and SiMPl^a_N and SiMPl^a_C (SiMPl^a), to use with ampicillin, respectively (Supplementary Fig. 1a, b). We confirmed that cells contained both SiMPl plasmids by analyzing the extracted plasmid DNA on a gel (Fig. 4i) and that the enzymes were reconstituted via Western blotting (Supplementary Fig. 3). We also compared the transformation efficiency of no plasmid, only pBAD33, only pTrc99a, both of these or SiMPl^c and SiMPl^a (Fig. 4j). SiMPl^c gave rise to 1.5 times more colonies than the co-transformation of pBAD33 and pTrc99a. SiMPl^a, instead, gave rise to several orders of magnitude fewer colonies compared to all other conditions. When titrating the amount of DNA used in the transformation, comparing pBAD33 and SiMPl^c we found no difference for all the concentrations used but one (0.1 ng), for which there were 1.6 times fewer colonies for SiMPl^c (Supplementary Fig. 4b). However, as stated above for SiMPl^k, it is important to note that we are comparing cells transformed with a single plasmid (pBAD33) with cells transformed with two plasmids (SiMPl^c). For SiMPl^a, already with 10 ng of DNA there were several orders of magnitude fewer colonies compared to pTrc99a (Supplementary Fig. 4c). For 1 ng of DNA there were only few colonies, while for 0.1 ng no colonies. Moreover, for all DNA concentrations below 100 ng, colonies took about 6 h longer to appear. Surprisingly, when growing cells in liquid culture, SiMPl^a performed just as well as pTrc99a (Supplementary Fig. 8).

**SiMPl for selection with hygromycin and puromycin.** The antibiotics that can be used to impose selective pressure to create

stable mammalian cell lines differ from those functional in bacteria. To make SiMPl applicable to mammalian cells, we therefore decided to engineer split versions of hygromycin B phosphotransferase and puromycin acetyltransferase (for simplicity, in the following referred to as HPT and PAT, respectively). Importantly, hygromycin B could be employed in plants as well. Despite having little sequence similarity, HPT and APT are structurally similar (Supplementary Fig. 9). We thought of taking advantage of this similarity to place the splice site in HPT in the same location—at the structural level—as the splice site in APT. We identified the splice site E105:T106 in HPT. However, this site did not support bacterial growth on hygromycin-selective plates. We therefore applied the protein structure flexibility analysis to HPT (Fig. 5a), as done with the other enzymes. From this analysis, two flexible sites were selected and tested (Fig. 5b). Of these, only one site (E255:L256) allowed the appearance of colonies on hygromycin-selective plates when both plasmids were transformed (Fig. 5c). To confirm that our splice site selection in flexible regions based on RMSF analysis was sound, we additionally intentionally tested two sites in regions of low flexibility, E219:A220 and D224:S225, expecting them to be dysfunctional (Supplementary Fig. 10a, b). Neither supported the formation of colonies on hygromycin-selective plates. When we looked at the location of these sites in the protein structure, we realized that both sites are in close proximity to strongly conserved residues and one directly contacts the ligand (Supplementary Fig. 10b). Weak sequence conservation, however, did not always allow identifying a functional splice site. For instance, splice site A155: D156, which lies in the weakly conserved region with the highest

flexibility (Supplementary Fig. 5 and Supplementary Fig. 10c, d) did not support bacterial growth on hygromycin-selective plates.

We used the only functional splice site (E255:L256) to construct pSiMPl$^h$_N and pSiMPl$^h$_C for use with hygromycin (Supplementary Fig. 1c). We confirmed the presence of both SiMPl plasmids in the bacteria by plasmid DNA extraction followed by analysis on an agarose gel (Fig. 5d) and protein *trans*-splicing via Western blotting (Supplementary Fig. 11). We further applied the mutations to gp41-1 and concluded that the activity of split HPT was entirely dependent on successful protein *trans*-splicing (Supplementary Fig. 12a). Sequencing showed that the N-terminal construct had one unexpected mutation (Fig. 5e). Evidently, the bacteria introduced this mutation in order to survive in the presence of hygromycin. To gain a mechanistic understanding of the role of this mutation, we first generated an ensemble of the protein model structure of the reconstituted enzyme. Then we computationally studied the area of the ligand (hygromycin) entry pocket in the wild type and the reconstituted enzyme, without and with the additional mutation (Supplementary Fig. 13). Interestingly, our analysis of the enzyme structure ensemble showed that the mutation that spontaneously occurred in the bacteria makes the area of the ligand entry pocket larger. The reconstituted enzyme without such mutation instead likely has a smaller entry pocket area for the ligand. Taken together, the data indicate that a functional splice site may be directly evolved in bacteria, as effectively the presence of an antibiotic represents a selective pressure.

To split PAT we first had to build a structural model of the protein, as there is no crystal structure available. To this aim, we used the RaptorX web server[22] using histone acetyltransferase as template (Supplementary Fig. 14). Using this predicted structure, we performed RMSF-based structure flexibility analysis (Fig. 5f). In this case, there were no obvious regions with higher flexibility compared to rest of the protein structure. We nonetheless selected four splice sites complying with the criteria for a good splice site (surface exposure, avoidance of the active site, low sequence conservation) (Fig. 5g). Of these, only one (V82:E83) supported formation of colonies on puromycin-selective plates when bacteria were transformed with both plasmids (Fig. 5h). We used this splice site to construct pSiMPl$^p$_N and pSiMPl$^p$_C for use with puromycin (Supplementary Fig. 2a). Presence of both plasmids was confirmed by agarose gel electrophoretic analysis of the DNA extracted from two randomly picked colonies (Fig. 5i). Successful enzyme reconstitution was assessed via Western blotting (Supplementary Fig. 3). For split PAT, full-length protein reconstitution was unnecessary, however the presence of gp41-1 was, since without the intein bacteria did not grow (Supplementary Fig. 12b).

**SiMPl is advantageous to produce valuable chemicals.** Many microbial metabolic engineering endeavors face the challenge to minimize the competition between pathways needed for cell growth and production of the desired product. One way to achieve this compromise between growth and production is to fine-tune the expression levels of several metabolic enzymes using inducible promoters[23] or other regulatory mechanisms[24]. We reasoned that diminishing the burden imposed on bacterial cells by cutting down the number of antibiotics to half could also help increase production of desired compounds. We first tested if, and to which extent, having to counteract two antibiotics instead of one affected bacterial growth. Under the tested conditions, the growth of *E. coli* TOP10 cells co-transformed with SiMPl$^k$ was only mildly better than that of cells co-transformed with pBAD33 and pTrc99a or SiMPl$^a$ (Supplementary Fig. 15). Cells co-transformed with SiMPl$^c$ grew less well than all others. We

nonetheless went on to test if we could achieve higher yields of some useful product with SiMPl$^k$ instead of two conventional plasmids.

As a first proof of principle, we decided to let *E. coli* cells produce indigoidine[25–27], a non-ribosomal peptide (NRP) with antimicrobial and antioxidant activities, which is also a blue pigment that can be conveniently quantified (Fig. 6a). Indigoidine is produced in the bacterium *Streptomyces lavandulae* by the NRP synthetase (NRPS) BpsA, which comprises an adenylation (A) domain, into which an oxidation (Ox) domain is embedded, a peptidyl carrier protein (PCP) domain and a thioesterase (TE) domain (Fig. 6b). BpsA needs to be first activated by an external phosphopantetheinyl transferase (PPtase), which attaches a PPant arm onto the PCP domain[28]. The substrate for indigoidine production is the amino acid glutamine. Indigoidine is a dimer of two oxidized and cyclized glutamines (Fig. 6b). The *E. coli* BAP1 strain[29], a derivative of BL21(DE3) with the promiscuous PPtase Sfp[30] integrated in its genome, produces indigoidine when expression of BpsA and Sfp is induced with IPTG (Fig. 6a). Interestingly, the role of the TE domain in the biosynthesis of indigoidine is not yet clear. Indeed, according to the proposed pathway, the TE domain would actually be dispensable. To clarify its role in the production of indigoidine as well as to test if the SiMPl method is advantageous to obtain higher levels of indigoidine, we eliminated the TE domain from the enzyme, thus generating BpsAΔTE (Fig. 6c). We then cloned the truncated enzyme and its excised TE domain each on a separate plasmid, either with a conventional resistance cassette or with the split cassette of SiMPl$^k$_N and SiMPl$^k$_C (Fig. 6d and Supplementary Fig. 16a–d). We additionally constructed a single plasmid bearing the conventional kanamycin resistance cassette containing BpsAΔTE and its TE domain in two separate MCSs (Fig. 6e and Supplementary Fig. 17). We then compared production of indigoidine from BAP1 cells transformed with the single, conventional plasmid, the two conventional plasmids or SiMPl$^k$. Cells exposed to a single antibiotic, regardless of whether carrying one or two plasmids, were able to produce ~29% more indigoidine than cells exposed to two antibiotics (Fig. 6f).

Next we applied the SiMPl method to a more complex metabolic pathway for the production of the industrially relevant non-proteinogenic aromatic amino acid *para*-amino-L-phenyla-lanine (L-PAPA)[31–33]. L-PAPA can be produced in *E. coli* by overexpression of three enzymes, PapA/PabAB, PapB and PapC, from organisms such as *Streptomyces venezuelae*[34], *Corynebacter-ium glutamicum*[35] or *Pseudomonas fluorescens*[33]. Higher yields of L-PAPA can be achieved when overexpressing additionally three enzymes from *E. coli* that increase the flux from glucose to chorismate through the shikimate pathway, namely the phospho-2-dehydro-3-deoxyheptonate aldolase (DAHP synthase) AroF, the 3-dehydroquinate synthase AroB and the shikimate kinase AroL[35] (Fig. 7a). We constructed the SiMPl$^k$ version of plasmids pJNT-*aroFBL* (pJNT_SiMPL$^k$_N) and pC53BC (pC53BC_SiMPl$^k$_C) (Fig. 7b and Supplementary Fig. 18) which were previously used to produce L-PAPA in the *E. coli* FUS4.7 R strain[35]. This strain is auxotroph for L-phenylalanine and L-tyrosine, thus it grows only when these amino acids are supplied. In this experimental setup, therefore, the potential advantage of SiMPl compared to two conventional plasmids would not lie in the growth of the cells, which is arrested during L-PAPA production. We first performed fed-batch shake flask cultivation and found that FUS4.7 R cells co-transformed with pJNT_SiMPl$^k$_N and pC53BC_SiMPl$^k$_C produced ~24% more L-PAPA than the same cells co-transformed with pJNT-*aroFBL* and pC53BC carrying conventional resistance cassettes (Fig. 7c). Finally, we tested production of L-PAPA in a small-scale bioreactor (Fig. 7d). Also in this case, we obtained more

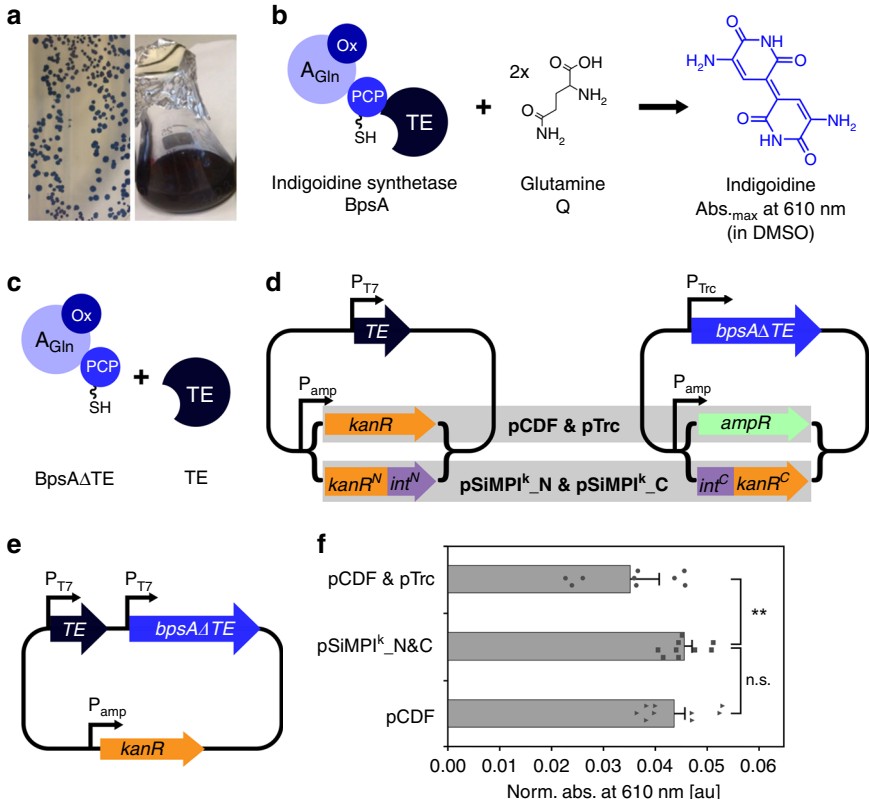

**Fig. 6** SiMPl plasmids increase production of the non-ribosomal peptide indigoidine. **a** Images of BAP1 cells producing indigoidine on plate (left) and in liquid culture (right). **b** Schematic of indigoidine biosynthesis. $A_{Gln}$, Adenylation (A) domain specific for glutamine; Ox, oxidation domain embedded within the A domain; PCP, peptidyl carrier protein domain. The PCP domain is depicted with the PPant arm attached to it; TE, thioesterase domain. **c** Schematic representation of the truncated BpsA (BpsA$\Delta$TE) and the externally supplied TE domain used in this study. **d**, **e** Schematic of the plasmids used in (**f**). **f** Bar graph showing the absorbance at 610 nm of overnight expression cultures in 85% DMSO of BAP1 cells expressing the indicated constructs from the indicated plasmids induced with 100 μM IPTG at 18 °C. Values were normalized to the value of the control (full-length BpsA expressed from pTrc99a) and represent mean (± standard error of the mean) of three biological replicates. ns, not significant (two-tailed, homoscedastic Student's t test).; double asterisk (**), p-value < 0.01 (two-tailed, heteroscedastic Student's *t* test). pCDF, plasmid shown in (**e**). Source data are provided as a Source Data file

(~15%) L-PAPA for cells co-transformed with the SiMPl plasmids (Fig. 7e).

**SiMPl can be used to enrich populations of mammalian cells.** We then tested if the SiMPl method could be applied to select mammalian cells double positive for the blue fluorescent protein mTagBFP2[36] and the green fluorescent protein ZsGreen1[37] using only puromycin. We cloned the *mtagbfp2* and *zsgreen1* genes into the SiMPl$^P$ lentiviral vectors (Supplementary Fig. 2b). Here the resistance cassette is driven by the spleen focus forming virus (SFFV) promoter, which is functional in mammalian cells[38] (Fig. 8a). We produced the viruses and transduced the human T cell line Jurkat with different amounts of either one virus or both viruses simultaneously. Untransduced cells were used as a control (Fig. 8b). All virus concentrations led to 100% transduced cells (Supplementary Fig. 19). To mimic the more realistic and typical situation in which the virus transduces only a fraction of cells, we mixed untransduced cells with cells transduced with one or two viruses in a 1:1:1:1 ratio. We obtained a cell mixture in which only about 25% of the cells express both parts of the split PAT and should survive selection using puromycin. The cells were cultured in medium without or with puromycin. As expected, the untransduced cells did not express the fluorescent proteins (Fig. 8c and Supplementary Fig. 20) and died in the presence of puromycin (Supplementary Fig. 21a). Upon selection, cells transduced with the SiMPl lentiviral vectors, which were alive as

assessed by flow cytometry (Supplementary Fig. 21a), were 100% positive for both fluorescent proteins (Fig. 8c and Supplementary Fig. 20). In contrast, the mixed population of cells kept on medium without selection contained only a small fraction of double positive cells (approximately 25% for 5 mL of virus) (Fig. 8c and Supplementary Fig. 20). These results suggest that only the presence of both SiMPl vectors gave rise to a functional PAT, whose correct reconstitution was confirmed by Western blotting (Supplementary Fig. 22); the individual fragments did not possess enzyme activity, in line with what observed in bacteria. For all virus amounts tested, it was possible to enrich cells double positive for both mTagBFP2 and zsgreen1 (Fig. 8c and Supplementary Fig. 20).

Finally, we applied SiMPl to select Jurkat cells expressing a functional T cell receptor (TCR) of murine origin on their surface. The TCR, employed by immune cells to combat pathogens or tumors, comprises eight subunits (TCRα, TCRβ and six CD3 subunits[39]) which are all required for its assembly and transport to the cell surface[40]. Jurkat cells lacking TCRα and TCRβ, therefore, do not express a TCR on their surface. We cloned the cDNAs encoding murine TCRα and TCRβ into the SiMPl$^P$ lentiviral vectors (Fig. 9a and Supplementary Fig. 23). We produced and concentrated the viruses and transduced cells with different amounts of either only one concentrated virus or both. Untransduced cells were used as a control (Fig. 9b). Transduced cells were 100% positive for the murine TCR (Supplementary

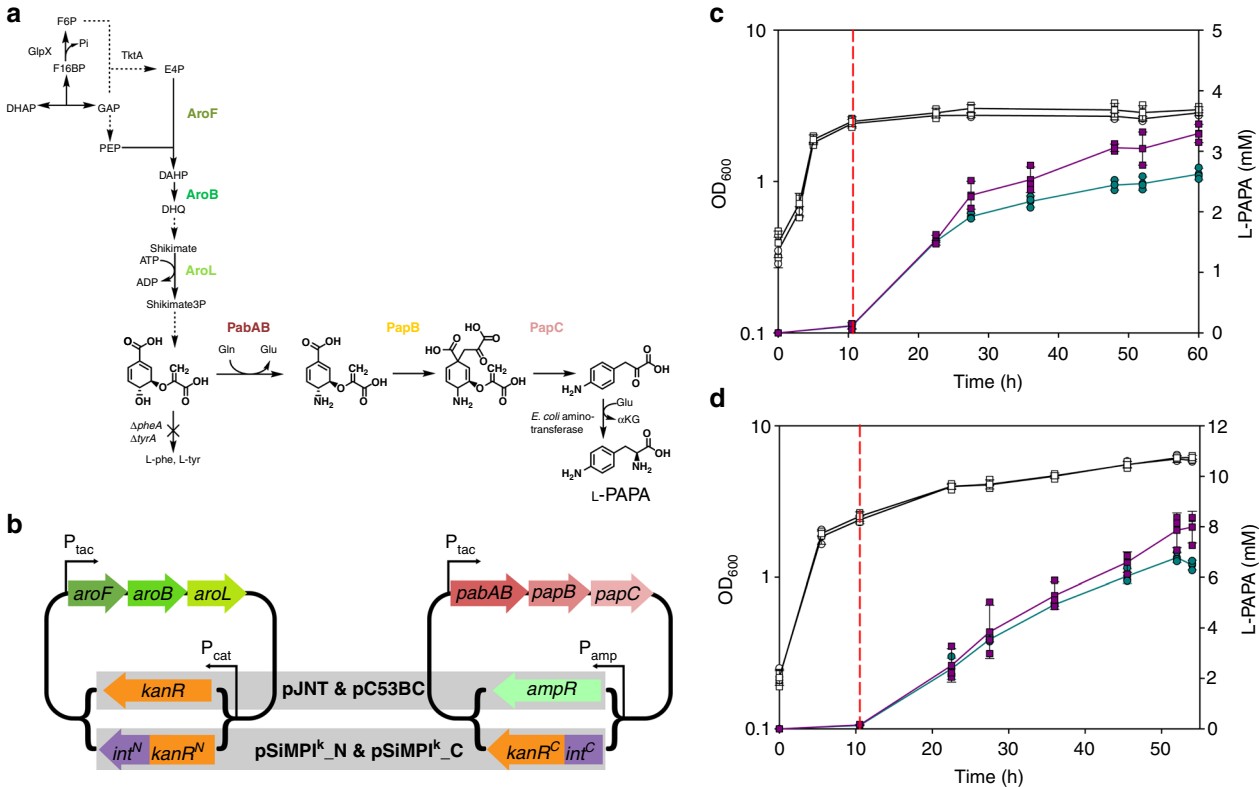

**Fig. 7** SiMPl plasmids increase production of the non-proteinogenic aromatic amino acid L-PAPA. **a** Schematic showing the de novo L-PAPA biosynthesis pathway in *E. coli*. Broken arrows indicate incomplete presentation of the metabolic pathway. Plasmid-borne overexpressed enzymes are colored. AroF, DAHP synthase; AroB, dehydroquinate synthase; AroL, shikimate kinase; GlpX, fructose-1,6-bisphosphate phosphatase; PabAB, 4-amino-4-deoxychorismate synthase; PapB, 4-amino-4-deoxychorismate mutase; PapC, 4-amino-4-deoxyprephenate dehydrogenase; PheA, bifunctional chorismate mutase/ prephenate dehydratase; TktA, transketolase A; TyrA, bifunctional chorismate mutase/ prephenate dehydrogenase. **b** Schematic showing the plasmids used in (**c**) and (**d**). **c, d** L-PAPA production in *E. coli* FUS4.7 R cells cultivated in flasks (**c**) or in a bioreactor (**d**). Optical density (OD$_{600}$) is shown in white. L-PAPA concentration in violet and green. Circles, production from cells co-transformed with pC53BC and pJNT-*aroFBL*; Squares, production from cells co-transformed with pC53BC_SiMPl$^k$_C and pJNT- SiMPL$^k$_N; Vertical dotted line, time at which production is initiated by addition of IPTG (0.5 mM final concentration). Values represent mean (± standard deviation) of three independent experiments. Source data are provided as a Source Data file

Fig. 24a). To mimic the situation in which the virus transduces only a fraction of cells and to exclude activity of the individual enzyme fragments, we mixed cells transduced with both SiMPl vectors simultaneously with cells transduced with only one SiMPl vector in a 1:1:1 ratio. We obtained a population of cells in which about 33% should express both parts of split PAT and thus survive puromycin selection. The cells were cultured in medium with or without puromycin. As expected, the untransduced cells as well as cells transduced with only one virus did not express the murine TCRβ chain on their surface and died in the presence of puromycin (Supplementary Fig. 21b and Supplementary Fig. 24b, c). Upon selection, cells transduced with the SiMPl lentiviral vectors, which were alive as assessed by flow cytometry (Supplementary Fig. 21b), were 100% positive for both chains of the murine TCR, as assessed by staining the surface of the cells with an antibody specific for murine TCRβ (Fig. 9c and Supplementary Fig. 25a; note that TCRβ is transported to the cell surface when TCRα is also expressed). In contrast, the mixed population of cells kept on medium without puromycin contained only 25% of cells with the murine TCRβ chain on the cell surface, confirming that the individual fragments of the enzyme did not possess activity. For all virus amounts tested, it was possible to enrich Jurkat cells double positive for both chains of the murine TCR using a single antibiotic (Fig. 9c and Supplementary Fig. 25a). The cells were functional as assessed by measuring calcium influx into the cytosol upon TCR stimulation (Fig. 9d).

One advantage of this enzyme splitting strategy is that the size of the resistance cassette decreases, an important feature when wanting to use, instead of lentiviruses, adeno-associated viruses (AAVs), which pose severe restrictions to the size of the transgene[41,42]. In order to allow more flexibility in the selection of PAT fragments of suitable size, we further identified four splice sites using our computational approach (Fig. 10a). Of these, two (V37:D38 and W191:C192) are close to the termini of the protein, giving rise to one small enzyme fragment. We cloned these variants into the SiMPl lentiviral vectors encoding the murine TCRα or TCRβ chains (Fig. 9a), and followed the same experimental procedure detailed above for the splice site V82:E83.

Untransduced cells did not survive selection with puromycin while cells transduced with the SiMPl lentiviral vectors encoding the new split PAT variants contained living cells as measured by flow cytometry (Supplementary Fig. 21c). All new splice sites enabled enrichment of cells positive for the murine TCR (Fig. 10b and Supplementary Fig. 25b), albeit for the splice site W191:C192 we had to cultivate the cells for longer to increase the number of living cells (Supplementary Fig. 26).

## Discussion

We have presented here SiMPl, a method based on intein-mediated protein *trans*-splicing that allows selection of cells containing two plasmids by using a single antibiotic at the same concentration used in the conventional approach based on

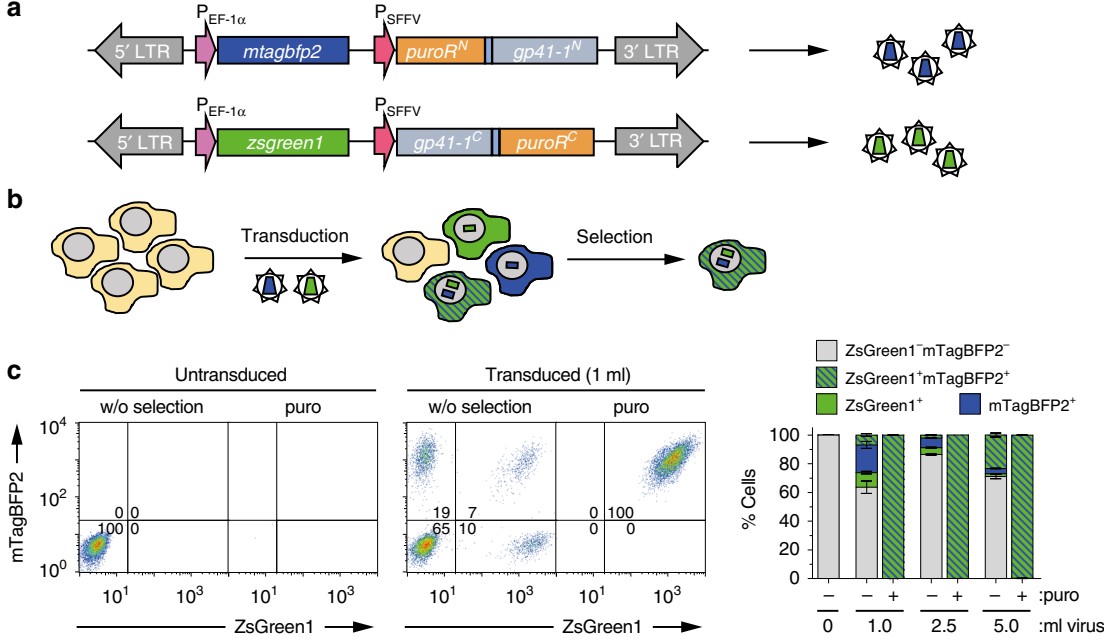

**Fig. 8** SiMPl can be used to select human T cells with puromycin. **a** Schematic of the SiMPl lentiviral vectors encoding mTagBFP2 and ZsGreen1 where the puromycin acetyltransferase is split at position V82:E83. **b** Schematic of the workflow. **c** T cells selected on puromycin after transduction with the two lentiviruses in (**a**) are 100% positive for mTagBFP2 and ZsGreen1. Left panel, representative flow cytometric analysis of mTagBFP2 and ZsGreen1 expression in living cells gated using forward versus side scatter. The numbers indicate the percentage of mTagBFP2- and/or ZsGreen1-positive cells in the indicated quadrants. Right panel, stacked bar chart showing the percentage of cells in the four quadrants (lower left, ZsGreen1$^-$ mTagBFP2$^-$; lower right, ZsGreen1$^+$ mTagBFP2$^-$; upper left, ZsGreen1$^-$ mTagBFP2$^+$; upper right, ZsGreen1$^+$ mTagBFP2$^+$) at the indicated conditions. Values represent mean (± standard deviation) of three independent experiments. Standard bar graphs with individual data points are shown in Supplementary Fig. 20. Source data are provided as a Source Data file

full-length enzymes. SiMPl reduces costs and adds convenience to the selection procedure. Moreover, cells are fitter when subjected to a single antibiotic. This has profound implications for instance for metabolic engineering endeavors, as healthier cells can produce larger amounts of valuable chemicals, as showcased here for the NRP indigoidine and the non-proteinogenic aromatic amino acid L-PAPA. Reducing toxicity could as well be advantageous when working with primary cells that cannot be cultured for too long.

Another advantage of SiMPl is that it enables the creation of complex synthetic circuits based on multiple parts. Indeed, the number of useful antibiotics to choose from is limited and their effectiveness is not identical[13,14]. Thus being able to select cells containing two vectors with only one antibiotic represents a strong asset. By combining SiMPl based on puromycin and hygromycin, it will be possible to select mammalian cells transduced with four different lentiviral constructs. While double, triple or even quadruple positive cells can also be enriched using fluorescence-activated cell sorting (FACS), dispensing all together of the selection pressure, the fluorescent channels would be unavailable for downstream analyses. In bacteria, the number of plasmids that can be co-transformed is limited by the availability of compatible origins of replication. Thus, even if theoretically the simultaneous use of SiMPl$^k$, SiMPl$^c$ and SiMPl$^a$ is feasible, the bottleneck remains the availability of only three compatible origins.

We decided to use a split intein to reconstitute the full-length enzyme in the cells, assuming that this would be more efficient than simply bringing the two enzyme fragments in physical proximity. However, we found that in some cases intein-mediated protein *trans*-splicing was dispensable. Nonetheless, the association between the intein fragments was always

necessary, playing the same role as other interacting pairs such as leucine zippers.

To find splice sites we adopted a computational approach based on the assumption that regions of high flexibility and low evolutionary sequence conservation would best tolerate the introduction of the local exteins' scar. The results indicate that indeed either protein flexibility or low conservation is a good predictor for the success of a splice site. However, interestingly, considering both high flexibility and low conservation did not always yield a functional splice site, as exemplified by splice site E87:Q88 for TEM-1. Finding working splice sites is overall challenging as splitting a protein in two dysfunctional parts may be associated with folding and solubility problems, which may be aggravated by fusion to poorly folding split intein fragments.

The approach described here is applicable to other enzymes and could be extended for selection in other organisms, such as plants, yeast or bacterial species in which there are few working antibiotics. We expect SiMPl to become a widely applied method whenever several plasmids are required.

## Methods

**Plasmid construction.** To clone EGFP into pBAD33, the *egfp* gene was amplified with a forward primer containing the *SacI* restriction site followed by a ribosome binding site (RBS) and a start (ATG) codon. The reverse primer had a stop codon and a restriction site for *HindIII*. The complete list of primers used in the present study is given in Supplementary Table 1. PCR amplification was carried out with Phusion Flash High-Fidelity PCR Master Mix (2x) from ThermoScientific using Biometra-Thermocycler from analytikjena. The double digested EGFP fragment was ligated into pBAD33 previously digested with *SacI* and *HindIII*. To remove the CAT gene from pBAD33_EGFP, the circular backbone was linearized by PCR omitting the resistance gene. To clone mRuby3 into pTrc99a, the *mruby3* gene was amplified by PCR with forward and reverse primers containing *EcoRI* and *HindIII* restriction sites, respectively. The double digested mRuby3 fragment was then ligated into pTrc99a previously digested using the same restriction enzymes. The

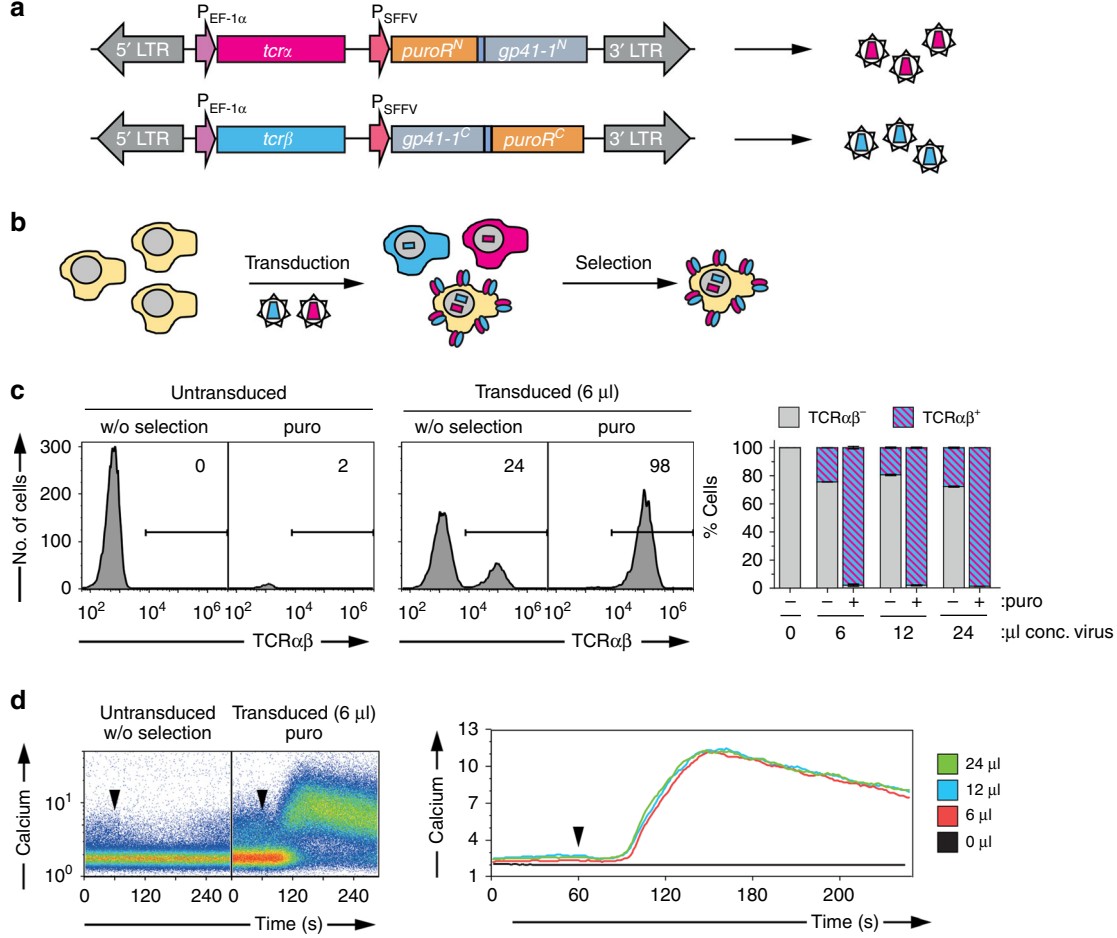

**Fig. 9** SiMPl allows selecting functional Jurkat T cells expressing a murine TCR on their surface. **a** Schematic of the SiMPl lentiviral vectors encoding TCRα and TCRβ where the puromycin acetyltransferase is split at position V82:E83. **b** Schematic of the workflow. **c** TCRα⁻TCRβ⁻ Jurkat T cells selected on puromycin after transduction with the two lentiviruses in (**a**) are 100% positive for the TCR. Left panel, representative histograms of living cells gated using forward versus side scatter and propidium iodide staining. Numbers indicate the percentage of cells expressing murine TCRβ on their surface (TCRαβ indicates that TCRα must be expressed to allow presentation of TCRβ on the cell surface). Right panel, stacked bar chart showing the percentage of TCRαβ⁺ cells for the indicated conditions. Values represent mean (± standard deviation) of three independent experiments. Standard bar graphs with individual data points are shown in Supplementary Fig. 25a. **d** Selected cells are functional. Intracellular calcium levels after stimulation with anti-murine TCRβ antibodies for the indicated cells. Left panel, dot plots of the calcium response depicted as ratio of Fluo-3 to Fura Red fluorescence over time. Right panel, overlay of the calcium responses of puromycin selected cells transduced with 6, 12 or 24 μL of concentrated lentivirus. The graph shows the 70th percentile of the ratio of Fluo-3 to Fura Red fluorescence over time. Source data are provided as a Source Data file

TEM-1 β-lactamase gene conferring resistance towards ampicillin was removed from the pTrc99a_mRuby3 circular backbone as described above for the chloramphenicol resistance gene. The N- and C- parts of the split kanamycin resistance cassette (containing the gp41-1 intein fragments) were amplified individually. Each part was finally assembled with the backbone using Gibson Assembly®. All other SiMPl plasmids were cloned using the same strategy. Plasmids were sequenced at Eurofins genomics.

For studying indigoidine production in *E. coli* cells, the *mruby3* gene in SiMPl^k_C was replaced with the DNA sequence coding for the TE domain of BpsA, while the *egfp* gene in SiMPl^k_N was replaced with the DNA sequence coding for BpsAΔTE using Gibson Assembly®. BpsA was codon optimized for expression in *E. coli* and kindly provided by the Fussenegger lab (Department of Biosystems Science and Engineering, ETH Zürich). Conventional plasmids, namely pTrc99a and pCDFDuet, containing BpsAΔTE and/or the separate TE domain, were assembled from PCR products featuring the required overlaps via Gibson Assembly®. The spectinomycin resistance gene (Sm^R) in the pCDFDuet vector was replaced by the kanamycin resistance gene (APT = kanR) from pET28a.

Lentiviral vectors containing split puromycin acetyltransferase were constructed using plasmid p526 (System Biosciences, Palo Alto, CA, USA) as backbone. The promoter in p526 was exchanged with a short version of the EF-1α (human elongation factor-1α) promoter. pLVX-S plasmids carrying the T1 TCR chains were used as a template to amplify by PCR the *tcrα* and *tcrβ* genes[43]. The gene downstream of the EF-1α promoter was replaced with *tcrα* in one plasmid and with

*tcrβ* in another plasmid. The *dsred2* gene downstream of the SFFV (spleen focus-forming virus) promoter was replaced with either the N-terminal part of the puromycin resistance gene followed by the N-terminal gp41-1 intein fragment or with the C-terminal gp41-1 intein fragment followed by the C-terminal part of the puromycin resistance gene. Individual fragments were PCR-amplified using Phusion Flash polymerase and assembled using Gibson Assembly®. The *mtagbfp2* and *zsgreen1* genes were synthetized and cloned into the SiMPl lentiviral plasmids pSiMPl^lp_N and pSiMPl^lp_C via Gibson Assembly®.

pC53BC_SiMPl^k_C and pJNT- SiMPl^k_N were constructed via Gibson Assembly® by swapping the full-length resistance genes in pC53BC and pJNT-*aroFBL* (whose construction is detailed in[35]) with the resistance cassettes of SiMPl^k_C and SiMPl^k_N, respectively. All PCR fragments were generated using Q5 DNA polymerase (NEB, USA). They were treated with *DpnI* (NEB, USA) and purified using a DNA purification kit (Clean & Concentrator-5, Zymo Research, USA). The constructed plasmids were verified by sequencing (Eurofins Genomics, Germany).

**Bacterial selection with hygromycin and puromycin.** E. coli TOP10 cells (Thermo Fisher Scientific C4040-03) were grown on low-salt LB agar-plates containing 100 μg/mL hygromycin. For puromycin, we used low-salt LB agar-plates adjusted to pH 8 using Tris with a final concentration of 50 mM containing 50 μg/mL puromycin.

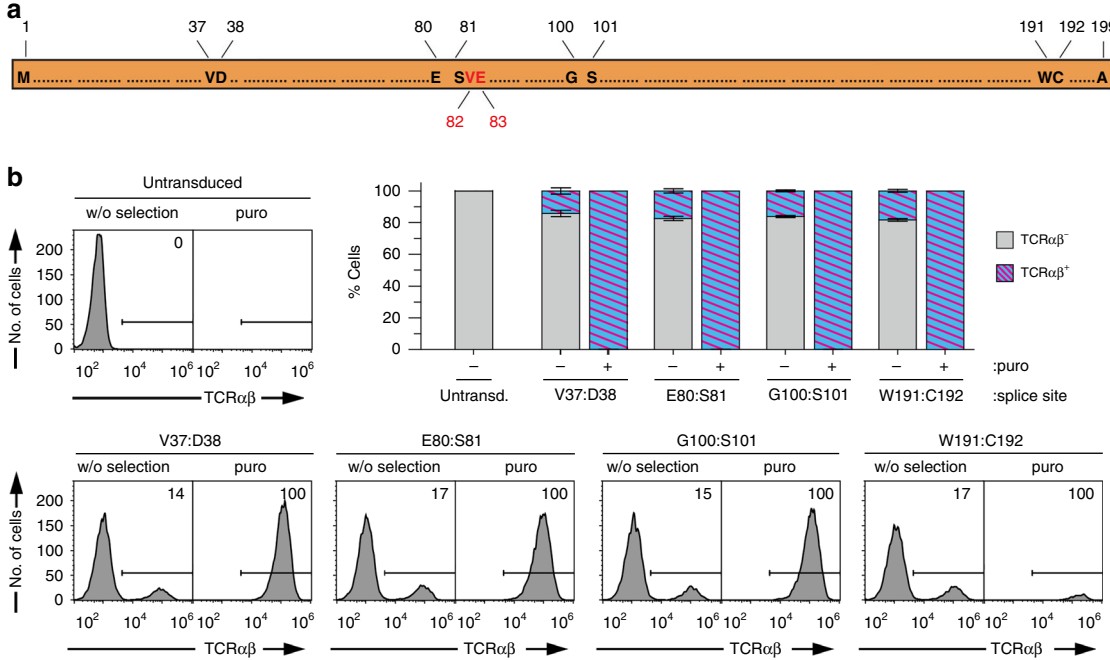

**Fig. 10** Characterization of further splice sites for puromycin acetyltransferase in human T cells. **a** Schematic representation of puromycin acetyltransferase where splice sites, and first and last amino acids are indicated. Remaining amino acids are shown as dots. Numbers represent the amino acid position. The first identified splice site V82:E83 is highlighted in red. E80:S81, G100:S101 and N192:C192 are scarless constructs, while V37:D38, V82:E83 and W191: C192 contain a scar of six residues ('SGY' at positions −3, −2, −1 and 'SSS', at positions +1, +2, +3). **b** Flow cytometric analysis of Jurkat T cells transduced with the indicated constructs. Upper left and lower panel, representative histograms of living cells gated using forward versus side scatter. Numbers indicate the percentage of murine TCRβ[+] cells. Upper right panel, stacked bar chart showing the percentage of murine TCRβ[+] cells under the indicated conditions. Values represent mean (± standard deviation) of three independent experiments. Standard bar graphs with individual data points are shown in Supplementary Fig. 25b. Source data are provided as a Source Data file

**Bacterial growth analysis**. For Supplementary Fig. 8, growth curves of *E. coli* TOP10 cells were obtained using the BioTek Synergy™ H4 plate reader. Briefly, 120 μL of nutrient broth (LB) with and without antibiotic(s) were administered into the wells of a flat-bottom 96-well plate. The overnight bacterial cultures were used to inoculate fresh cultures into the 96-well plate with an optical density ($OD_{600}$) of 0.01. To avoid evaporation of the medium during measurements, the plate was closed with a lid and the sides were wrapped with parafilm tape. The reader was driven by Gen5 v2.01.14 software (temperature = 37 °C, run time = 20 h, read interval = 2 min and 43 s, wavelength = 600 nm, shake = slow, shake once every 130 s, read = absorbance end point, read speed = normal and delay = 100 msec). The reader was preheated before the start of the experiment. Chloramphenicol, ampicillin and kanamycin were used at a final concentration of 35 μg/mL, 100 μg/mL and 50 μg/mL, respectively. For Supplementary Fig. 15, growth curves were measured in 100 mL conical flasks containing 20 mL of cultures in LB with the corresponding antibiotics incubated at 37 °C with shaking (220 rpm). Cultures were started with an $OD_{600}$ of 0.1 and were grown until $OD_{600} = 0.5$ after which 0.1% arabinose and 100 μM IPTG were added. Absorbance was measured at regular intervals using $OD_{600}$ DiluPhotometer™ (IMPLEN).

**Analysis of protein reconstitution in bacteria**. Overnight cultures were used to inoculate day cultures starting at an $OD_{600} = 0.1$ which were grown at 37 °C 250 rpm until $OD_{600} \sim 1$. Subsequently, 200 μL of each culture were pelleted and resuspended in 10 μL 4x Laemmli buffer (Bio-Rad) by vortexing. The tubes were heated at 95 °C for 10 min and stored at −20 °C. PAGE separation was performed using Mini-PROTEAN® TGX™ Precast Gels (Bio-Rad) at 120 V for ~1 h. Proteins were then transferred to PVDF membranes using Trans-Blot® Turbo™ Transfer Packs and a Trans-Blot® Turbo™ Transfer System (Bio-Rad). The membranes were placed in 4–5% BSA dissolved in TBST and incubated with agitation for 2 h at room temperature. Then they were rinsed with TBST and incubated with rat anti-HA (1:2000, Cat# 11867423001, Sigma), rabbit anti-FLAG (1:2000, Cat# AHP1074, Bio-Rad), and mouse anti-GAPDH (1:1000, Cat# G13-61M, SignalChem) primary antibodies for 2 h at room temperature. Three 5-minute washes with TBST were followed by incubation with AlexaFluor 488-conjugated anti-rat (1:2000, Cat# A-11006, Invitrogen), Cyanine5-conjugated anti-rabbit (1:2000, Cat# A10523, Invitrogen), and AlexaFluor 790-conjugated anti-mouse (1:2000, Cat# A28182, Invitrogen) secondary antibodies for 1 h at room temperature. After washing the

membrane twice with TBST for 5 min, fluorescent signals were measured by an Amersham Typhoon imaging system (GE Healthcare) using 488, 635 and 785 nm wavelength lasers.

**Protein Cα fluctuation analysis**. Crystal structures of chloramphenicol acetyltransferase (PDB ID: 1q23), hygromycin B phosphotransferase (PDB ID: 3w0s) and TEM-1 beta-lactamase (PDB ID: 1zg4) were downloaded from the RCSB PDB database[44]. The structure of puromycin acetyltransferase was modeled via the RaptorX web-server[22] using histone acetyltransferase as template (PDB ID: 2qec chain A, 29.8% sequence similarity). Fluctuation (flexibility) analysis of the Cα carbon atoms in the protein backbone was studied using the CABS-flex 2.0 web server with the default parameters[20].

**Computational analysis of hygromycin B phosphotransferase**. The structures of the scar-containing WT and mutated reconstituted hygromycin B phosphotransferase were modeled using the SWISS-MODEL web-server[45]. The CABS-flex web-server[20] was used to obtain near-native structural dynamics of all three constructs (WT without scar, WT with scar and mutant with scar). Ten representative protein structures of the CABS-flex simulation outputs for the WT enzyme without scar, with scar and the mutant enzyme with scar were analyzed in PyMOL. A triangle obtained by the Cα atoms of Gly32, Pro135 and Arg236 was determined for each snapshot and the ligand entry area was calculated.

**Evolutionary trace analysis**. Evolutionary trace analysis (ETA) was performed online on the ETA web server[46] by submitting Uniprot IDs of the proteins.

**Indigoidine expression and quantification**. BAP1 cells[29] (kind gift of Blaine A. Pfeifer) were grown in LB medium supplemented with the appropriate antibiotic/s (final concentrations: 100 μg/mL (ampicillin), 50 μg/mL (kanamycin)) at 180 rpm and 37 °C overnight. These starter cultures were diluted to an $OD_{600}$ of 0.05 in M9 minimal medium (1x M9 salts (M9-Minimal salts 5x, powder, SERVA, Heidelberg, Germany), 1 mM $MgSO_4$, 0.5% casamino acids (OmniPur®, Merck, Darmstadt, Germany), 0.4% glucose). The expression cultures were grown in 14 mL polypropylene tubes (Greiner Bio-One, Frickenhausen, Austria) at 37 °C and 180 rpm to an $OD_{600}$ of about 0.6–0.8, cooled down at 4 °C for 5 min and induced

with 100 nM IPTG (isopropyl β-D-1-thiogalactopyranoside, Sigma-Aldrich, St. Louis, Missouri, USA). The expression proceeded overnight at 18 °C and 180 rpm.

The absorption of 15 µL overnight expression culture in 85% DMSO was measured in a clear, flat-bottom 96-well plate (PlateOne, Greiner Bio-One) using a plate reader (TECAN, Männedorf, Switzerland). The relative amount of blue pigment was calculated according to Myers et al.[47]. In short, to assess the amount of pigment independently from the cell density, we measured the absorption of 15 µL overnight expression culture in 85 µL DMSO in a clear 96 well plate in the TECAN plate reader at both, 600 and 800 nm. The ratio δ between $OD_{600}$ and $OD_{800}$ of a negative control sample was calculated and used to infer the relative amount of indigoidine in the samples of interest as follows:

$$\text{relative pigment production of construct } X = OD600_X - (\delta * OD800_X)$$

**Experiments with TCRα⁻TCRβ⁻ Jurkat T cells.** $10 \times 10^6$ or $2.5 \times 10^6$ Human embryonic kidney cells (HEK-293T, ATCC: CRL-11268) were transfected with 19.5 µg or 7 µg DNA, respectively, consisting of the lentiviral vectors and the packaging and envelope plasmids (pCMVΔR8.74 and pMD2.vsvG, kind gift of Didier Trono) in a 2:1:1 ratio using polyethyleneimine (Polysciences). Lentiviral supernatant was collected 24 h and 48 h after transfection, centrifuged and filtered through a 0.45 µm pore size filter. The lentiviral supernatants were directly used for transduction or virus concentration was performed by centrifugation for 4 h at 10,000g through 10% (w/v) sucrose, 50 mM Tris (pH7.4), 100 mM NaCl and 0.5 mM EDTA. Supernatants were discarded and the viral pellet was resuspended in 100 µL PBS. The amount of lentiviral supernatant or concentrated lentivirus used to transduce $0.5 \times 10^6$ TCRα⁻TCRβ⁻ Jurkat T cells (Clone E6-1 (ATCC® TIB-152™) genetically modified to delete *tcrα* and *tcrβ*) is indicated in the figures. Cells transduced with the same amount of individual or both viruses of the SiMPl vectors encoding ZsGreen1 and mTagBFP2 (Fig. 8a) were mixed with untransduced cells in a 1:1:1:1 ratio prior selection with puromycin. Cells transduced with the same amount of individual or both viruses of the SiMPl$^p$ vectors encoding TCRα and TCRβ (Fig. 9a) were mixed in a 1:1:1 ratio prior selection with puromycin. Cells were divided into triplicates and cultured in medium without or with addition of puromycin. Cells transduced with the SiMPl vectors encoding ZsGreen1 and mTagBFP2 (Fig. 8, Supplementary Fig. 21a) were selected for one day with 0.3 µg/mL puromycin (puro) and additional 6 days with 0.6 µg/mL puromycin and analysis was performed 7 days after the removal of puromycin from the culture medium. Cells transduced with the SiMPl vectors encoding TCRα and TCRβ (Fig. 9, Supplementary Fig. 21b) were selected with 0.5 µg/mL puromycin for 4 days and puromycin containing medium was replaced by standard medium one day before the analysis. Cells transduced with the SiMPl vectors encoding TCRα and TCRβ and the new split PAT variants (Fig. 10, Supplementary Fig. 21c) were selected with 0.6 µg/mL puromycin for 7 days and cultured for additional 6 days in medium without puromycin prior analysis. Cells transduced with the SiMPl vectors encoding TCRα and TCRβ and the HA- or flag-tagged PAT or spliced PAT (Supplementary Fig. 22) were selected with 0.6 µg/mL puromycin for 5 days and analysis was performed 7 days after the removal of puromycin from the culture medium. Untransduced cells served as a control. For flow cytometric analysis of murine TCRβ expression, cells were washed once with PBS containing 2% fetal calf serum (FCS) and resuspended in 20 µL of a 100-fold dilution of allophycocyanin-conjugated anti-murine TCRβ (clone H57-597, eBioscience) in PBS containing 2% FCS. Cells were stained for 20 min on ice and washed twice with PBS containing 2% FCS. When indicated, 5 µM propidium iodide was added to the cell suspension just before flow cytometric analysis. Cells were measured using the Gallios or CyAn ADP flow cytometer (Beckman Coulter) and analyzed using the FlowJo software (Tree Star). Cell lines were not tested for the presence of *Mycoplasma*.

**Measurement of Ca²⁺ influx.** TCRα⁻TCRβ⁻ Jurkat T cells were left untransduced or lentivirally transduced with the SiMPl$^p$ vectors (Fig. 8a), and selected with 0.5 µg/mL puromycin for 5 days. $1 \times 10^6$ cells were resuspended in 200 µL RPMI 1640 medium supplemented with 10% FCS, 0.1% pluronic F-127 (Invitrogen), 2.6 µM Fluo-3 AM (Life Technologies), and 5.5 µM Fura Red AM (Invitrogen). Cells were loaded in the dark with the calcium dyes for 45 min at 37 °C, washed and kept on ice until measurement. For analysis, cells were diluted in pre-warmed medium and maintained at 37 °C during the measurement with the CyAn ADP flow cytometer (Beckman Coulter). Baseline calcium levels were measured for 1 min, then 5 µg/mL anti-mouse TCRβ (clone H57-597, homemade) was added as indicated. The stimulation was recorded for further 4 min. Data were analyzed with the FlowJo software (Tree Star).

**Analysis of protein reconstitution in Jurkat T cells.** $50 \times 10^6$ TCRα⁻TCRβ⁻ Jurkat T cells, lentivirally transduced with the SiMPl$^p$ vectors (Supplementary Fig. 23), were lysed in 1 mL 20 mM Tris (pH = 7.4), 137 mM NaCl, 10% (v/v) glycerol, 2 mM EDTA, 1 mM PMSF, 5 mM IAA, 10 mM NaF, 0.5 mM Na3VO4, protease inhibitor cocktail (Sigma-Aldrich) and 0.5% (v/v) Brij96V for 30 min at 4 °C. The lysate was cleared by centrifugation for 15 min at 18,500g and mixed with 1 µg anti-FLAG (clone M2, Sigma-Aldrich) or anti-HA (clone 12CA5, homemade) and protein G sepharose beads (GE Healthcare). Immunoprecipitation was performed overnight at 4 °C. Following SDS-PAGE separation, immunoblotting was

performed using anti-FLAG (rabbit polyclonal, Bio-Rad), anti-HA (clone 114-2C-7, Merck Millipore) or anti-GAPDH (rabbit polyclonal, Sigma-Aldrich) antibodies.

**Shake flask experiments for L-PAPA production.** Electrocompetent *E. coli* FUS4.7 R cells were electroporated with either plasmids pC53BC and pJNT-aroFBL or plasmids pC53BC_SiMPl$^k$_C and pJNT- SiMPl$^k$_N. The resulting two strains were first cultivated in 250 mL shake flasks with 25 mL LB for 8 h in a shaking incubator at 37 °C and at 200 rpm agitation. 0.2 mL of this culture were transferred to 10 mL minimal medium (MM; 3 g/L $KH_2PO_4$, 12 g/L $K_2HPO_4$, 5 g/L $(NH_4)_2SO_4$, 0.3 g/L $MgSO_4 \cdot 7H_2O$, 0.1 g/L NaCl, 0.1125 g/L $FeSO_4 \cdot 7H_2O$/Na citrate 15 mL (from the solution of 7.5 g/L $FeSO_4$ and 100 g/L trisodium citrate), 0.015 g/L $CaCl_2 \cdot 2H_2O$, 1 g/L yeast extract, 7.5 µg/L thiamine HCl, 0.04 mg/mL L-phenylalanine, 0.04/mg mL L-tyrosine and 5 g/L glucose) in a 100 mL shake flask to be cultured for 14 h. 0.25 mL of the seed culture were used to inoculate a 250 mL shake flask containing 25 mL MM. IPTG was added to a final concentration of 0.5 mM after 10.5 h to enable the expression of the plasmid-borne enzymes and additional 5 g/L glucose was added and the temperature was shifted to 30 °C. Ampicillin sodium salt (100 mg/L) and/ or kanamycin sulfate (50 mg/L) was/were added when appropriate. Samples were taken frequently and the supernatants were isolated and stored at −20 °C until high-performance liquid chromatography (HPLC) analysis.

**Bioreactor experiments for L-PAPA production.** Fed-batch cultivations were performed in a 0.75 L benchtop bioreactor (Multifors 2, Infors HT Switzerland) with a batch volume of 380 mL MM and a final volume of 430 mL. As described above for the flask cultivation, 4 mL of the seed culture were used to inoculate the bioreactor. The operating temperature was 37 °C, the pH of 7.0 was regulated by the addition of 10% ammonia solution and the partial oxygen concentration (pO2) was maintained above 30% saturation by adjusting stirrer speed and aeration rate. After 10.5 h cultivation IPTG (final concentration 0.5 mM), 0.04 mg/mL L-phenylalanine and 0.04 mg/mL L-tyrosine were added to the medium to induce the expression of the plasmid-borne enzymes and to form additional biomass. Further, the temperature was shifted from 37 °C to 30 °C and a glucose feed was started to maintain a concentration above 5 mM. To ensure complete consumption of 90 mM glucose, the cells were cultivated for 54 h. Samples were taken frequently and the supernatants were isolated and stored at − 20 °C until HPLC analysis.

**Determination of L-PAPA and glucose concentrations.** For the determination of L-PAPA and glucose concentration, supernatants at given time points were isolated by centrifugation (21,000g, 10 min). The samples were stored at −20 °C until HPLC analysis. This was performed on the 1260 Infinity HPLC system (Agilent Technologies). L-PAPA was detected at 210 nm with a diode array detector (DAD, 1260 Infinity, Agilent Technologies). For separation, a Prontosil C18 column (250 × 4 mm, CS-Chromatographie Services, Germany) was used at 40 °C. As a mobile phase, 40 mM $Na_2SO_4$ (pH 2.7 adjusted with methane sulfonic acid) was used with a flow rate of 1 mL/min. L-PAPA was quantified using 1.0 mM L-phenylglycine as the internal standard to correct variabilities in analytes and a five-point calibration curve with L-PAPA as an external standard. Glucose quantification was performed by HPLC with a refractive index detector (RID; 1260 Infinity, Agilent Technologies). An Organic Acid column (300 × 8 mm, CS-Chromatographie Services, Germany) was used at 40 °C with 5 mM $H_2SO_4$ and a flow rate of 0.8 mL/min. Glucose was quantified using a five-point calibration curve with glucose as an external standard.

**Reporting summary.** Further information on research design is available in the Nature Research Reporting Summary linked to this article.

## Data availability
The SiMPl plasmids for use in bacteria have been deposited at Addgene with the following IDs: 134310 (pSiMPl$^k$_N and pSiMPl$^k$_C); 134312 (pSiMPl$^c$_N and pSiMPl$^c$_C); 134314 (pSiMPl$^a$_N and pSiMPl$^a$_C); 134316 (pSiMPl$^h$_N and pSiMPl$^h$_C); and 134318 (pSiMPl$^p$_N and pSiMPl$^p$_C). The SiMPl lentiviral vectors are available upon request. For all figures (main and supplementary) containing experimental data, the raw data were compiled in a single excel file available with this manuscript as Source Data file. Any other relevant data are available from the authors upon reasonable request.

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

## Acknowledgements

The authors thank Simone L. von Löwensprung for support with lentivirus production, Kersrtin Fehrenbach for growing the T cells and Hanna Wetzel for support with initial cloning. This work was funded by the Deutsche Forschungsgemeinschaft (DFG) under Germany's Excellence Strategy through EXC294 (BIOSS—Center for Biological Signal-ling Studies), EXC2189 (CIBSS—Centre for Integrative Biological Signalling Studies, Project ID 390939984) and GSC-4 (Spemann Graduate School, A.M.). B.D.V. was additionally financed by the DFG (grant VE 776/3-1 within the SPP1926). G.S. and J.Y. were financed by the Ministerium für Wissenschaft und Kunst Baden-Württemberg.

## Author contribution

N.P. and B.D.V. conceived the study; N.P. and M.A.Ö. performed computational ana-lyses; N.P. performed all cloning and bacterial experiments except the experiments with indigoidine and the Western blotting under the supervision of B.D.V.; A.D. performed experiments with indigoidine under the supervision of B.D.V.; J.B.B. performed Western blotting from bacterial samples under the supervision of B.D.V.; A.M., C.J. and W.S. performed experiments with T cells; J.Y. performed experiments on L-PAPA production; J.Y. and G.S. analyzed data on L-PAPA production; G.S., W.S. and B.D.V. secured funding; B.D.V. wrote the paper with input from all authors.

## Competing interests

The authors declare no competing interests.
