## [Peer Review File · Nature Communications]

Reviewers' Comments:

Reviewer #1:

Remarks to the Author:

In the current manuscript by Di Ventura and coworkers, the authors describe a split intein-based strategy for selection of bacterial or mammalian cells bearing two plasmids with just a single antibiotic. Below are comments for their consideration.

1. As it is currently presented, Figure 2 presents only qualitative information about bacterial transformation. The authors should report the transformation efficiency in terms of CFU per microgram of DNA for the different experiments reported in Figs. 2. It would also be helpful to see how this compares to the plasmids (pBAD33 and pTrc99) individually and when co-transformed. And have the authors varied the DNA concentration to see how this impacts transformation efficiency?
2. The experiment documented in Fig. 3b is missing two important control cases. Specifically, the authors need to include the number of colonies that arise from transformation with just pBAD33 alone or just pTrc99 alone to determine how many colonies would arise from each of these plasmids individually.
3. There are many literature reports of split beta-lactamase (Galarneau et al. 2002 Nat Biotechnol; Wehrman et al., 2002 PNAS), which render false the authors' statement that this enzyme has never been split before (line 138). It is also curious that the authors didn't test these well known split sites for their intein-mediated designs. In a similar vein, Albert Cheng and coworkers have recently reported split hygromycin and split puromycin selectable markers (Jillette et al. bioRxiv 2019).
4. The only evidence provided that the split CAT and split AMP systems actually worked for colony selection is a DNA gel showing digestion of isolated plasmid DNA. These newly designed systems need to be much more rigorously characterized including the addition of quantitative information on transformation efficiency and how this compares to the single plasmid controls. The authors should also similarly determine whether there is any advantage conferred by their CAT and AMP constructs over the traditional 2-plasmid co-transformation as they did for the split Kan system in Fig. 3b.
5. Using empty plasmids to compare to pSiMPI plasmids that encode EGFP and mRuby seems like a poorly designed control. Shouldn't plasmids all encode same/similar genes since the presence/absence of these will impact cell growth. Especially if the encoded genes are leaky.
6. The impact of the different pSiMPI constructs on growth as documented in Fig. 6a is questionable as all the curves are nearly overlapping (and there are no error bars or statistical analysis). The authors should tone down any growth advantages that they claim stem from reducing the antibiotic load as this seems to be a marginal effect.
7. The details of the metabolic engineering application (i.e., indigoidine production) are poorly described, with details of the different plasmids completely absent from the results, methods and figure caption. This makes it impossible to determine what has actually been accomplished in this experiment. Also, given such a simple system, it's not clear that the split enzyme method would even be needed. Instead, one could simply clone both gene products/fragments into a single plasmid with little difficulty. In fact, such a control should be included. The authors might also consider a better demonstration of the utility of their system as the one provided does not effectively highlight any advantages of their system over conventional methods.
8. While the TCR demonstration is interesting and convincing, the authors have skipped the important step of demonstrating/characterizing the baseline transfection of mammalian cells with the split plasmid system using GFP/mRuby as they did for the bacterial constructs. This is an important missing piece to the current manuscript and needs to be included so that readers can judge the efficacy of the approach before seeing it used in an application.
9. In several places, the authors claim "data not shown," which is no longer an acceptable practice. These results should be added to the manuscript or supplement, or else not mentioned.
Minor issues
10. Figure 1 is rather low quality from an aesthetics standpoint and should be visually improved if

possible.

11. The phrase “yielded way fewer colonies” (line 129) is rather informal language and lacks any sufficient quantitative information. Could the authors define/state the numbers instead? In general, this is illustrative of the overall lack of quantitation throughout the manuscript.

Reviewer #2:

Remarks to the Author:

In the manuscript, Navaneethan et al. showed an intein-mediated selection approach called SiMPI in both bacteria and mammalian cells for selecting two plasmids. They split the resistance gene including ampicillin, chloramphenicol, kanamycin, hygromycin, and puromycin and applied the split marker for selecting double positive expression of TCR subunits and producing indigoidine in bacteria. However as said in the manuscript, the split-intein based strategy is a common protein split strategy. The authors should better clarify the major advances over previous studies. A specialized journal would be more suitable for this manuscript.

Major issues

1. The approaches adopted in the manuscript is less convincing with many “data not shown” in the manuscript. These missing data sets should be clarified and included. In general, the results are not sufficient to justify their conclusions and the authors should provide more data to support their claims and the advantage of their methods.
2. The intein-mediated self-catalytic splicing reaction usually requires specific amino acids residue at the +1 position downstream of the C-terminal intein fragment. It is necessary to adeexplain how authors split the resistance gene. The data is not compared with a suitable control, such as split without intein. And no direct proof was provided to show that the intein worked in a trans-splicing that linked the two split fragments.
3. The co-selecting approache for integrating TCR two chains into Jurkat cells is interesting. However, the results are less convincing. It is necessary to demonstrate the function of the selected Jurkat cells and whether both of TCR alpha and beta are expressed.
4. As the split marker would result in lower expression of the marker genes, authors should test the protein level of the reconstituted protein and assembly efficiency in both bacteria and mammalian cells. The indirect evaluation by the colonies and OD is not convincing. Do the split markers require a higher level of the drug to reach the sufficient selection in bacteria and mammalian cells?
5. One advantage of the split marker system is that it may save the payload space, especially in mammalian lentivirus/AAV delivery. Thus, the authors should test more split sites and compare their efficiency, protein expression level and the impact on the selective drug.

Minor issues,

1. In line 52, the description “several years ago” should be avoided.
2. In Figure 2c, the data should be displayed by exact colony number and corresponding error bar.

Reviewers' comments:

Reviewer #1 (Remarks to the Author):

In the current manuscript by Di Ventura and coworkers, the authors describe a split intein-based strategy for selection of bacterial or mammalian cells bearing two plasmids with just a single antibiotic. Below are comments for their consideration.

We thank the reviewer for the many insightful comments. Our reply to each point can be found below.

1. As it is currently presented, Figure 2 presents only qualitative information about bacterial transformation. The authors should report the transformation efficiency in terms of CFU per microgram of DNA for the different experiments reported in Figs. 2. It would also be helpful to see how this compares to the plasmids (pBAD33 and pTrc99) individually and when co-transformed. And have the authors varied the DNA concentration to see how this impacts transformation efficiency?

In the new Fig. 2c, 3c, 4e-h, 5c and 5h we now show the transformation efficiency of the SiMPI plasmids in *E. coli* TOP10 cells. In Fig. 4j, we now compare the efficiency of transformation for pBAD33 alone, pTrc99a alone, both of these plasmids together, SiMPI^a and SiMPI^c. In Fig. 3c we compare the transformation efficiency of SiMPI^k, pBAD33+pTrc99a and pET28a. We additionally varied the amount of DNA and show the transformation efficiency obtained with 100, 10, 1 and 0.1 ng of DNA in new Suppl. Fig. 4.

2. The experiment documented in Fig. 3b is missing two important control cases. Specifically, the authors need to include the number of colonies that arise from transformation with just pBAD33 alone or just pTrc99 alone to determine how many colonies would arise from each of these plasmids individually.

As mentioned above in the reply to point 1, we now show this comparison in the new Fig. 4j. We show it there because we think it is more appropriate to compare pBAD33 (CAM) and pTrc99a (AMP) to the SiMPI^a (AMP) and SiMPI^c (CAM) plasmids, rather than SiMPI^k (KAN). In Fig. 3c, we compare SiMPI^k (KAN) to pET28a (KAN).

3. There are many literature reports of split beta-lactamase (Galarneau et al. 2002 *Nat Biotechnol*; Wehrman et al., 2002 *PNAS*), which render false the authors' statement that this enzyme has never been split before (line 138). It is also curious that the authors didn't test these well known split sites for their intein-mediated designs. In a similar vein, Albert Cheng and coworkers have recently reported split hygromycin and split puromycin selectable markers (Jillette et al. *bioRxiv* 2019).

We are grateful to the reviewer for pointing this to us. We sincerely apologize for having failed to identify the previous work on split beta-lactamase. We now cite the relevant papers and mention that this enzyme was split before. According to our computational approach, however, the previously selected residues where to split the protein (one being between G194 and L196 and one between E195 and L196; our numbering) would not be our first choice. Therefore, we did not use these sites in the current manuscript. We now mention this in the main text and show in the new Supplementary Fig. 6 the position of the previously established G194:L196 site in our RMSF plot.

We did not cite the *bioRxiv* paper by the Cheng group, instead, not because we were not aware of its existence, but because we have been in contact with them as soon as their paper was uploaded while we were preparing our own manuscript. We agreed on coordinating a back-to-back publication, because neither of us would like to undermine the work of the other. We would be uncomfortable to state "hygromycin B phosphotransferase and puromycin acetyltransferase have been split before", because when we worked on these enzymes they had not been split before! We would like to cite the paper by the Cheng group as a note at the end of the manuscript. We are happy to further discuss this issue with the editor.

4. The only evidence provided that the split CAT and split AMP systems actually worked for colony selection is a DNA gel showing digestion of isolated plasmid DNA. These newly designed systems need to be much more rigorously characterized including the addition of quantitative information on transformation efficiency and how this compares to the single plasmid controls. The authors should also similarly determine whether there is any advantage conferred by their CAT and AMP constructs over the traditional 2-plasmid co-transformation as they did for the split Kan system in Fig. 3b.

We have now measured transformation efficiencies (TE) for both SiMPI^a and SiMPI^c (new Fig. 4e-h). Moreover, we have compared the TE achieved with these to the ones achieved with only pBAD33, only pTrc99a or both of these traditional plasmids. These data are shown in new Fig. 4j.

5. Using empty plasmids to compare to pSiMPI plasmids that encode EGFP and mRuby seems like a poorly designed control. Shouldn't plasmids all encode same/similar genes since the presence/absence of these will impact cell growth. Especially if the encoded genes are leaky.

We agree with the reviewer. We now cloned the *egfp* and *mruby3* genes into pBAD33 and pTrc99a, respectively, and measured the growth curves of MG1655 cells transformed with no plasmid, pBAD33 (*egfp*) + pTrc99a (*mRuby3*) and SiMPI^{k/a/c}. The data are shown in new Supplementary Fig. 15.

6. The impact of the different pSiMPI constructs on growth as documented in Fig. 6a is questionable as all the curves are nearly overlapping (and there are no error bars or statistical analysis). The authors should tone down any growth advantages that they claim stem from reducing the antibiotic load as this seems to be a marginal effect.

We apologize for the poor quality of old Fig. 6a, where the error bars were extremely hard to detect. The reviewer is right in saying that the difference in the growth is minimal, with SiMPI^k being marginally better than pBAD33+pTrc99a, at least under the tested experimental conditions. We tuned down any claim about growth advantage in the main text accordingly.

7. The details of the metabolic engineering application (i.e., indigoidine production) are poorly described, with details of the different plasmids completely absent from the results, methods and figure caption. This makes it impossible to determine what has actually been accomplished in this experiment. Also, given such a simple system, it's not clear that the split enzyme method would even be needed. Instead, one could simply clone both gene products/fragments into a single plasmid with little difficulty. In fact, such a control should be included. The authors might also consider a better demonstration of the utility of their system as the one provided does not effectively highlight any advantages of their system over conventional methods.

This experiment was a proof of principle that cells exposed to a single antibiotic can produce higher amounts of a valuable chemical. The reviewer is right that indigoidine is made by a single enzyme which, albeit being very large (~ 3800 bp), fits on a single plasmid. We routinely use such a construct to produce indigoidine. We selected to work with indigoidine for two reasons: 1) because, being a pigment, it is easy to quantify; 2) because we were interested in elucidating the role of the TE domain in indigoidine synthesis and therefore wanted to express BpsAΔTE and the TE domain separately. We could have cloned both on a single plasmid, as noted by the reviewer (which we now did, see below). However, the plasmid we had in our lab with two multiple cloning sites (pCDF) has twice the T7 promoter. To study the relation between BpsAΔTE and its excised TE domain, we wished to separately control the expression levels of BpsAΔTE and the TE domain, thus for us it was natural to rather clone them into pBAD33 and pTrc99a. This further offered us the opportunity to compare production of indigoidine with the SiMPI^k pair of plasmids which were nothing but pBAD33 and pTrc99a modified purely in the resistance cassettes. Given the new data on the more complex metabolic pathway (see below) and the space restriction, we now do not show this data set anymore (old Fig. 6e; we are happy to include the data again in case you or the editor deemed it necessary). To address the reviewer's comment, we now added the control of the single plasmid carrying both components. Specifically, we first exchanged the spectinomycin resistance cassette originally contained on pCDF with a full-length kanamycin resistance cassette to be able to compare indigoidine production from one or two plasmids in cells exposed to the same antibiotic (kanamycin). Then we cloned into this modified pCDF (in the first multiple cloning site) the DNA sequence coding for the TE domain. Additionally, we created a SiMPI version of this plasmid by swapping the full-length kanamycin cassette with the split one. Finally, we also cloned in the second MCS the DNA sequence coding for BpsAΔTE. In the new Fig. 6f we show that the levels of indigoidine obtained with the single plasmid equal those obtained with SiMPI. Cells exposed to two antibiotics produce less indigoidine. We now show schematics of the plasmids used for this experiment in new Fig. 6d,e (in the first submission the plasmid maps were shown in Supplementary Fig. 9 only). We moreover show the detailed plasmid maps in new Supplementary Fig. 16 and 17.

Finally, as suggested by the reviewer, we applied SiMPI in a more complex metabolic pathway where six enzymes are overexpressed (in groups of three from two plasmids). Specifically, we show in the new Fig. 7 the production of the industrially relevant, non-proteinogenic aromatic

amino acid *para*-amino-L-phenylalanine (L-PAPA). We show that higher amounts of L-PAPA are obtained with the SiMPl plasmids using fed-batch flask cultivation as well as a bioreactor.

8. While the TCR demonstration is interesting and convincing, the authors have skipped the important step of demonstrating/characterizing the baseline transfection of mammalian cells with the split plasmid system using GFP/mRuby as they did for the bacterial constructs. This is an important missing piece to the current manuscript and needs to be included so that readers can judge the efficacy of the approach before seeing it used in an application.

We have performed this experiment, albeit using the green fluorescent protein ZsGreen1 and the blue fluorescent protein mTagBFP2 (blue). The cells that survived puromycin selection were double-positive for green and blue. The experiment is now shown in new Fig. 8.

9. In several places, the authors claim “data not shown,” which is no longer an acceptable practice. These results should be added to the manuscript or supplement, or else not mentioned.

We have eliminated any occurrence of “data not shown” and show the data in either the main text or the supplement. For dysfunctional splice sites, however, there is nothing to show because bacteria did not grow. In these cases, we now simply state this.

Minor issues

10. Figure 1 is rather low quality from an aesthetics standpoint and should be visually improved if possible.

Following the suggestion of the reviewer, we prepared an entirely new Fig. 1.

11. The phrase “yielded way fewer colonies” (line 129) is rather informal language and lacks any sufficient quantitative information. Could the authors define/state the numbers instead? In general, this is illustrative of the overall lack of quantitation throughout the manuscript.

We thank the reviewer for pushing us to be quantitative in this manuscript. We now corrected this and state precisely the numbers wherever suitable.

Reviewer #2 (Remarks to the Author):

In the manuscript, Navaneethan et al. showed an intein-mediated selection approach called SiMPI in both bacteria and mammalian cells for selecting two plasmids. They split the resistance gene including ampicillin, chloramphenicol, kanamycin, hygromycin, and puromycin and applied the split marker for selecting double positive expression of TCR subunits and producing indigoidine in bacteria. However as said in the manuscript, the split-intein based strategy is a common protein split strategy. The authors should better clarify the major advances over previous studies. A specialized journal would be more suitable for this manuscript.

We thank the reviewer for the feedback. Yes, inteins have been around for a long time; it is not a new technology *per se*. The application of inteins to reconstitute split enzymes conferring resistance towards antibiotics is, however, novel. We do cite the proof of principle *ACS Synthetic Biology* paper by Schmidt and colleagues where, for the first time, the idea of maintaining multiple plasmids with a single antibiotic was shown (here with leucine zippers and not intein-mediated protein reconstitution). However, as we explain in the manuscript, the implementation of the method was impractical, likely being the reason why it did not become wide-spread. Moreover, that method was confined to bacteria. Our work brings this technology a step further, allowing its use in mammalian cells and, in the future, in other organisms, such as plants. We do think this is a very important achievement, which will benefit a large number of scientists (which justifies, in our opinion, why this manuscript is suited for a journal of broad readership such as *Nature Communications*). As a matter of fact, puromycin is among the best working antibiotics for the selection of mammalian cells. We spoke to many colleagues in the mammalian synthetic biology field and received feedback that such a method, allowing selection of double positive cells with puromycin only, is highly desirable. We have now edited the main text to highlight what sets our work apart from previous ones and what the advantages of our methods are. After explaining the method by Schmidt and colleagues, we now write: *“This approach has one major drawback: the genes belonging to the circuitry conferring resistance (T7 polymerase and/or two enzyme fragments) are cloned under inducible promoters in the multiple cloning site (MCS). However, typically, it is desirable to employ the inducible promoters to control the expression of genes of interest rather than the components of the resistance cassette. This hindered the usage of the system for real applications. Moreover, the method was applied only to the previously split APT¹² and was limited to bacteria. The*

possibility to select plant and mammalian cells containing two plasmids with a single antibiotic would be beneficial for basic as well as applied science, because there are not many antibiotics that work well in these model systems^{13,14}.”

Moreover, our method has a strong computational component which sets it apart from most previously published papers on split enzymes.

Major issues

1. The approaches adopted in the manuscript is less convincing with many “data not shown” in the manuscript. These missing data sets should be clarified and included. In general, the results are not sufficient to justify their conclusions and the authors should provide more data to support their claims and the advantage of their methods.

We have now eliminated any occurrence of “data not shown” and show the data wherever possible (e.g., for dysfunctional splice sites cells did not grow thus there is nothing to show). We also provide more data for the readers to assess the performance of the SiMPI plasmids for the different antibiotics (new Fig. 4e-h, 4j, 5c, 5h and new Supplementary Fig. 4).

2. The intein-mediated self-catalytic splicing reaction usually requires specific amino acids residue at the +1 position downstream of the C-terminal intein fragment. It is necessary to adeexplain how authors split the resistance gene. The data is not compared with a suitable control, such as split without intein. And no direct proof was provided to show that the intein worked in a trans-splicing that linked the two split fragments.

We apologize for the lack of clarity in the previous version of the manuscript. We have now edited the figure (new Fig.1) and the text to clarify that gp41-1 has serine as catalytic residue at position +1. We moreover better explain that we decided to insert additional five residues beyond the serine (the so-called local exteins which are derived from gp41-1 natural host protein) to ensure high *trans*-splicing efficiency. We also overall changed the wording to present how we select splice sites. Actually, the fact that we do this following a rational approach, rather than a trial-and-error one, is one of the major features of this manuscript. We have a rather long paragraph detailing the criteria and the corresponding computational approaches we adopted (paragraph in the first section of Results starting with “To expand the SiMPI toolbox”).

Regarding showing a control: we mutated either the conserved cysteine – which is the very first residue of the N-terminal intein fragment – or the conserved asparagine – which is the very last residue of the C-terminal intein fragment. Since these residues are both necessary for the splicing reaction, these mutants tell us whether *trans*-splicing is required or not. These data were shown in the old Fig. 3a,b and old Supplementary Fig. 4. Now they are shown in new Fig. 3a,b and new Supplementary Fig. 7 and 12. We performed the “no intein” control only for puromycin acetyltransferase split at V82:E83 (shown in new Supplementary Fig. 12b), because this was the only case for which enzyme activity was unaffected by mutation to either the cysteine or the asparagine. The results show that here *trans*-splicing is not required but the intein fragments are needed to bring the enzyme parts in close physical proximity.

Regarding the direct proof of the occurred *trans*-splicing: we have now performed Western blotting for all our constructs which confirmed that full-length enzymes are present. We show the results in new Supplementary Figures 3, 11 and 21.

3. The co-selecting approach for integrating TCR two chains into Jurkat cells is interesting. However, the results are less convincing. It is necessary to demonstrate the function of the selected Jurkat cells and whether both of TCR alpha and beta are expressed.

We have now added functional data to show that the selected Jurkat T cells respond to activation of the TCR present on the cell surface with an influx of calcium (new Fig. 9d).

The TCR complex is built out of several subunits, including TCR α and TCR β . Already back in the 90s it was shown by many labs that, if one of these subunits is missing, e.g. either TCR α or TCR β , a TCR cannot be expressed on the cell surface. We have cited a review for that. The reason is that there is a strict quality control mechanism in the endoplasmic reticulum, and if the complex is not complete, wrongly assembled or partially misfolded, it will not be transported to the cell surface, but degraded instead. This is not specific to the TCR, but occurs with many protein complexes in the secretory pathway. Thus, we are confident that a surface expression of the TCR proves that both TCR α and TCR β were expressed. In the first submission, we had written this sentence to explain how we could be sure that both TCR subunits are expressed: “*Only when all subunits are present in the cell, a complete TCR is assembled and transported to the cell surface*³¹. Thus, we chose TCR α and TCR β -double deficient Jurkat cells (TCR α TCR β), which is a human T cell line that does not express a TCR on its surface, due to the lack of both TCR α and TCR β . [...] Upon selection, cells transduced

with the SiMPl lentiviral vectors, which were alive as assessed by flow cytometry analysis (Fig. 7c, upper panel), were almost 100% positive for both chains of the murine T1 TCR, assessed by staining of the cells with an antibody specific for murine TCR β (Fig. 7c, lower panel).”. We have now extensively re-written this paragraph (to accommodate new results also in response to reviewer #1 and to meet the word limit requirement) and now state: “The TCR, employed by immune cells to combat pathogens or tumours, comprises eight subunits (TCR α , TCR β and six CD3 subunits³⁹) which are all required for its assembly and transport to the cell surface⁴⁰. Jurkat cells lacking TCR α and TCR β , therefore, do not express a TCR on their surface.” and “Upon selection, cells transduced with the SiMPl lentiviral vectors, which were alive as assessed by flow cytometry (Supplementary Fig. 20b), were practically 100% positive for both chains of the murine TCR, as assessed by staining the surface of the cells with an antibody specific for murine TCR β (Fig. 9c; note that TCR β is transported to the cell surface when TCR α is also expressed).”

4. As the split marker would result in lower expression of the marker genes, authors should test the protein level of the reconstituted protein and assembly efficiency in both bacteria and mammalian cells. The indirect evaluation by the colonies and OD is not convincing. Do the split markers require a higher level of the drug to reach the sufficient selection in bacteria and mammalian cells?

We used the same concentration of antibiotics for selecting cells transformed/transduced with SiMPl vectors as for cells transformed/transduced with the traditional plasmids. Scientists can use SiMPl with the same plates/media that they would use normally. We have now added a sentence to highlight this in the Discussion.

We performed Western blotting to prove that the enzymes are reconstituted (new Supplementary Figures 3, 11 and 21). We would like to point out, however, that we are reluctant to speak of *trans*-splicing *efficiency*. In order to do so, one would have to possess information about the concentrations of both N- and C-intein constructs in the reaction. Only in this case, we would be able to state the percentage of product formed in respect to the amount that could be maximally made, that is, the same concentration of the least abundant between the N- and C-intein constructs. However, we cannot determine the concentration of N- and C-intein construct expressed independently, because cells do not grow when they receive only one of the two constructs. Lacking this information, we could nonetheless estimate efficiency by looking at how much N- and C-intein construct is left in the cells co-transformed/transduced

with both SiMPl plasmids. If one of the two constructs were not visible, but the spliced product were, we could conclude that it has been entirely consumed, thus the product has been made to its maximum amount. The efficiency would be 100%. We would be able to do this for most constructs for which we could clone tags at both termini without disrupting the enzyme's function; however, not for all constructs. Moreover, for chloramphenicol acetyltransferase and hygromycin B phosphotransferase, the N-terminal intein construct has the same size as the reconstituted enzyme, not allowing us to know how much of the N-construct is left. Therefore, because we cannot quantify efficiency for all constructs, we prefer not to talk of efficiency, but simply to state that enzyme reconstitution was confirmed by Western blotting.

5. One advantage of the split marker system is that it may save the payload space, especially in mammalian lentivirus/AAV delivery. Thus, the authors should test more split sites and compare their efficiency, protein expression level and the impact on the selective drug.

The reviewer is right that, especially for AAVs, the possibility to reduce the size of the transgene is beneficial and that more “asymmetrical” splice sites would be even better, given that one enzyme fragment would be much smaller. We have now identified and tested four additional splice sites, among which two that are close to the termini of the protein. We characterized all these new sites in terms of expression of the TCR on the cell surface (new Fig. 10). All but one were as efficient as the first identified site. Once again, we would like to point out that we use the standard puromycin concentration. We did not test different concentrations of the antibiotic because we explicitly wish to use the very same amount that would be normally used.

Regarding comparing the constructs at the protein level, we did not do this because we cannot control the virus titers and cells might get infected with different numbers of viral particles which may lead to different expression levels.

Minor issues,

1. In line 52, the description “several years ago” should be avoided.

We have removed it.

2. In Figure 2c, the data should be displayed by exact colony number and corresponding error bar.

We now show transformation efficiency for all constructs with error bars (new Fig. 2c, 3c, 4e-h, 5c and 5h).

Reviewers' Comments:

Reviewer #1:

Remarks to the Author:

The authors have adequately addressed all of the concerns that were raised previously.

Reviewer #2:

Remarks to the Author:

The authors have provided additional data to address most of my comments. However, I have to point out again that the idea of splitting proteins/markers has been published many times before, and the advantage to use the intein-mediated split biomarker selection over the multiple biomarker selection is still less convincing.

REVIEWERS' COMMENTS:

Reviewer #1 (Remarks to the Author):

The authors have adequately addressed all of the concerns that were raised previously.

We thank once again the reviewer for the many insightful comments and we are happy to read that our revision could address all their concerns.

Reviewer #2 (Remarks to the Author):

The authors have provided additional data to address most of my comments. However, I have to point out again that the idea of splitting proteins/markers has been published many times before, and the advantage to use the intein-mediated split biomarker selection over the multiple biomarker selection is still less convincing.

We regret that we did not manage to convince the reviewer of the novelty and power of SiMPL. We nonetheless are happy to read that our revised manuscript could address most of the comments the reviewer had.